# Dimension-Reduction Attack! Video Generative Models are Experts on Controllable Image Synthesis

**Hengyuan Cao**
Zhejiang University

**Yutong Feng**[†]
Kunbyte AI

**Biao Gong**
Ant Group

**Yijing Tian**
Hangzhou Normal University

**Yunhong Lu**
Zhejiang University

**Chuang Liu**
Hangzhou Normal University

**Bin Wang**
Kunbyte AI

{caohy, yunhonglu}@zju.edu.cn tianyijing2002@163.com liuchuang@hznu.edu.cn
{fengyutong.fyt, a.biao.gong, binwang393}@gmail.com

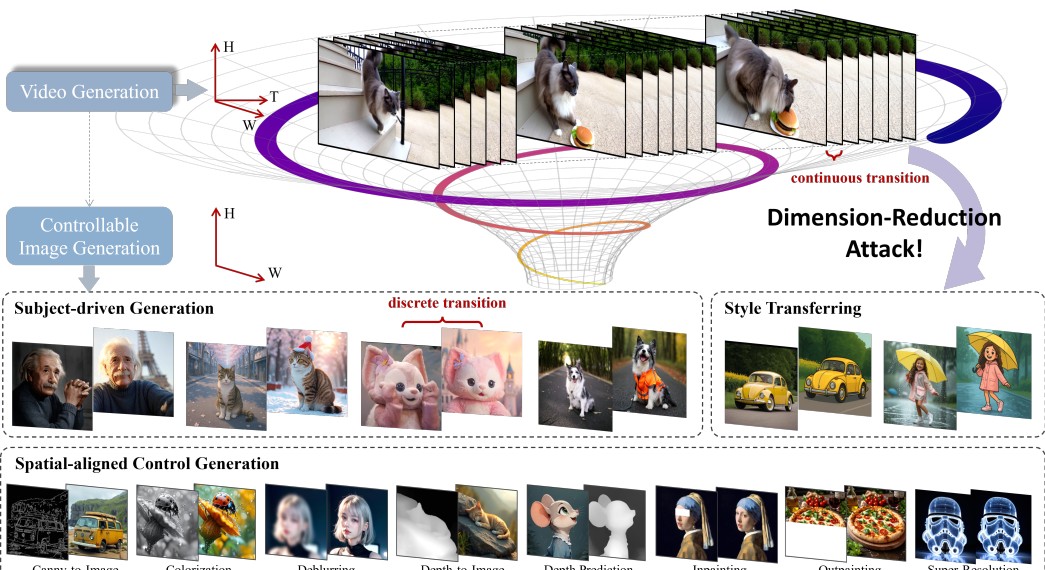

Figure 1: This paper leverages high-level prior of video generative models to unify controllable image generation in low-level. Bottom results show various types of task supported by `DRA-Ctrl`.

## Abstract

Video generative models can be regarded as world simulators due to their ability to capture dynamic, continuous changes inherent in real-world environments. These models integrate high-dimensional information across visual, temporal, spatial, and causal dimensions, enabling predictions of subjects in various status. A natural and valuable research direction is to explore whether a fully trained video generative model in high-dimensional space can effectively support lower-dimensional tasks such as controllable image generation. In this work, we propose a paradigm for video-to-image knowledge compression and task adaptation, termed *Dimension-Reduction Attack* (`DRA-Ctrl`), which utilizes the strengths of video models, including long-range context modeling and flatten full-attention, to perform various generation tasks. Specially, to address the challenging gap between

[†]Project leader.

39th Conference on Neural Information Processing Systems (NeurIPS 2025).

continuous video frames and discrete image generation, we introduce a mixup-based transition strategy that ensures smooth adaptation. Moreover, we redesign the attention structure with a tailored masking mechanism to better align text prompts with image-level control. Experiments across diverse image generation tasks, such as subject-driven and spatially conditioned generation, show that repurposed video models outperform those trained directly on images. These results highlight the untapped potential of large-scale video generators for broader visual applications. `DRA-Ctrl` provides new insights into reusing resource-intensive video models and lays foundation for future unified generative models across visual modalities. The project page is `https://dra-ctrl-2025.github.io/DRA-Ctrl/`.

# 1   Introduction

Recent advances in text-to-image (T2I) generative models [38, 34, 9, 23] have significantly improved the quality of image synthesis from natural language prompts. To enhance controllability, researchers have introduced auxiliary conditions into the context of generation [54, 24, 49, 32, 60, 10, 18], such as subject reference images, edge maps and depth cues. This has given rise to the paradigm of *controllable image generation*, where both textual and visual conditions collaboratively guide the synthesis process. While early methods relied on additional image adapters or cross-attention mechanisms [54, 6, 15, 42, 44], recent approaches leverage *full-attention* architectures [61, 22, 50, 43, 51, 7, 25, 12, 29, 47] that treat all input tokens as a unified sequence. However, these models are all built upon image generative models, thus remain limited by the static nature of image data, which lacks the continuous temporal and causal structures transformation present in the real world.

Video generative models [2, 53, 20, 45], in contrast, are trained to predict sequences of frames with rich spatiotemporal dependencies. The prior knowledge learned by these models incorporates long-range context, consistent object transitions, non-rigid transformation and high-level scene dynamics. These capabilities align closely with the goals of controllable image generation. This observation inspires a new direction, *i.e.,* repurposing pretrained video models to support image-level tasks by transferring their high-dimensional knowledge into a lower-dimensional setting. This work dives into this idea and presents a framework termed `DRA-Ctrl` that efficiently adapts video generators for diverse controllable image generation scenarios.

However, directly adapting video generative models to controllable image generation presents non-trivial challenges. A naive baseline would be to gather the condition image and target image into the frame sequence of video generators. The key hindrance confronted here is that the video data inherently consists of temporally continuous frames with smooth transitions, while the condition-target image pairs represent a discrete, abrupt change between two states. In detail, we investigate to adapt two variants of video generative models treating the image pairs as two-framed video. For image-to-video (I2V) model consuming the condition image as the first frame, it suffers to over-constrain the output to mimic the condition image. While for text-to-video (T2V) model, it is inevitable to inject the condition image as non-noisy frame tokens into the sequence. Thus the model takes much efforts to readapt the new paradigm, and tends to forget its pre-training knowledge with suboptimal performance. These baseline solutions expose the fundamental discrepancy between the continuous dynamics learned by video models and the discrete transition required by controllable image generation. Therefore, it is essential for `DRA-Ctrl` to conduct stable transferring when repurposing the video models without forgetting their high-dimensional capabilities.

To address these challenges in `DRA-Ctrl`, we propose a *mixup-based transition* strategy, inspired by the mixup [57] principle in representation learning, serving as a bridge connecting the diverse intermediate gaps in videos and images. The core idea is to treat the condition and target images as boundary frames of a synthetic shot transition sequence, with intermediate frames generated using a temporal position-aware mixup. Each intermediate frame is weighted by its relative position between the two endpoints, enabling smooth interpolation while preserving key visual characteristics. We implement the mixup transition with the I2V model. When integrating with these augmented frames, the constraint from condition to target images is significantly relaxed, making it easier to adapt to discrete image generation. Despite this, real video transitions generally require dozens of intermediate frames, resulting in dramatically increased computation cost. To mitigate this, we introduce *Frame-Skip Position Embedding*, a positional encoding scheme that expands temporal intervals in the latent space, allowing large image transformations with only a few frames. Additionally, to distinguish

complex combination of subjects and environments in multiple images, we adapt the condition and target prompts into the full-attention mechanism together with a masking strategy.

We evaluate `DRA-Ctrl` on a wide range of controllable image generation tasks, including subject-driven image synthesis, spatially aligned condition generation (*e.g.,* canny-to-image translation, colorization, deblurring, depth-based generation and depth prediction), masking image generation (inpainting and outpainting) and style transferring. Our experiments demonstrate that video generative models can be effectively re-purposed for these tasks, consistently outperforming methods built upon image generative models. This surprising effectiveness highlights a compelling "*Dimension-Reduction Attack*", where high-dimensional video priors offer enhanced control when adapted to lower-dimensional image tasks, encouraging more efforts to further investigate the extending capability of video generative models.

## 2   Related Works

**Subject-driven Image Generation.** Subject-driven image generation with diffusion models typically follows two paradigms: tuning-based and tuning-free methods. Tuning-based methods [40, 11, 16, 21] achieve strong identity consistency but require per-subject fine-tuning, limiting scalability and introducing non-trival computational overhead. Tuning-free methods instead enhance generalization through training on large-scale datasets, eliminating inference-time tuning. Early works [54, 24, 49, 32, 60, 18] extract subject information from reference images using an image encoder, and inject these features into the generation process via cross-attention mechanisms. Then, Hu et al. [15] propose using a ReferenceNet which is architecturally identical to the denoising UNet as the image feature extractor, providing detailed and accurate control information for controllable generation. Later advancements in tuning-free methods leverage the model's inherent in-context learning capabilities [17], treating the model itself as an image feature extractor to provide subject-specific information for generation. Zeng et al. [56] proposes to model the joint distribution of multiple text-image pairs sharing the same subject, investigating in-context learning within UNet-based diffusion models for subject-driven image generation. With the introduction of DiT architectures [33], recent works [61, 22, 50, 43, 51, 7, 25] have explored the full-attention mechanism, where reference images and generated images jointly participate in self-attention, to facilitate subject feature extraction and enable in-context learning for subject-driven generation. We propose leveraging video diffusion models' inherent frame-level full-attention mechanism for subject-driven generation.

**Spatially-aligned Image Generation.** Spatially-aligned control signals for fine-grained image generation have emerged as a critical research direction. Early conditional Generative Adversarial Networks (GANs) [19, 63] and transformers [4] achieve image-to-image translation by learning the mapping from conditional images to target images. Recent diffusion models enable tighter integration of such controls. SDEdit [30] guides generation process by first adding noise to stroke paintings and then denoising them. In contrast, T2I-Adapter [31] trains an adapter network to enable more diverse and precise control signals. ControlNet [58] reuses the encoding layers of pre-trained diffusion models as a backbone for learning control signals. UniControl [36] further advances this direction by integrating multiple tasks within a unified framework via a task-aware HyperNet, demonstrating zero-shot capabilities on unseen tasks and combined tasks. Subsequent works [7, 51, 43, 25, 12, 29, 47] have unified subject-driven and spatially-aligned image generation within one framework that maps control images to target outputs, which `DRA-Ctrl` also follows.

**Image Generation with Video Models.** While existing works employ video generative models for *image editing* (requiring pixel-aligned partial modifications) that are methodologically naive, our framework targets *controllable image generation* that enables comprehensive transformations — including background replacement, subject pose/state alteration, and holistic content regeneration. FramePainter [59] injects interactive editing signals extracted by the control encoder into the generation process via cross-attention mechanisms and synthesizes a two-frame video where the first frame reconstruct the condition image and the second one produces the edited output. Object-Mover [55] addresses the object relocation task by fine-tuning a video generative model through frame-wise concatenation of condition images with various control signals. Rotstein et al. [39] proposes a direct I2V approach for image editing, where condition images and Vision Language Model (VLM)-processed prompts are jointly fed into the model, with edited results obtained through a specialized frame selection strategy. While Lin et al. [27] and Chen et al. [7] similarly employ video models for controllable image generation or editing tasks, primarily motivated by their ability

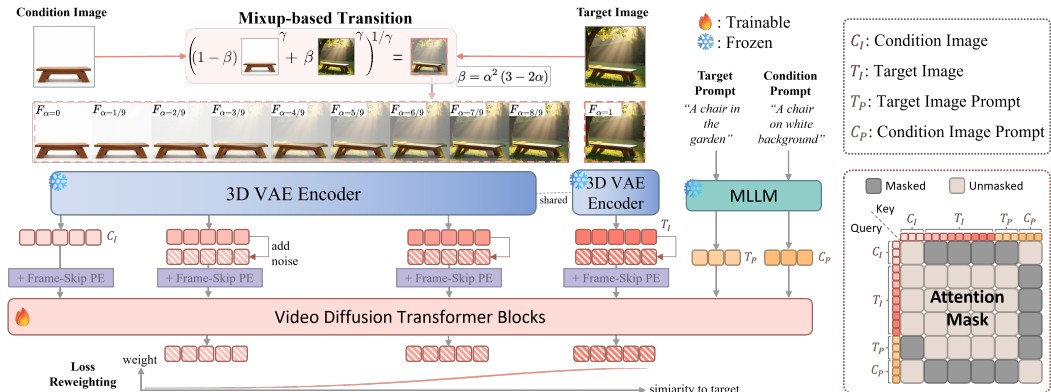

Figure 2: The **training framework** of `DRA-Ctrl`. We propose a mixup-based transition strategy to construction shot transition videos to adapt the video model for abrupt image changes, with FSPE strategically reducing transitional frames. The loss function is adaptively reweighted according to the proportion of target image in the token sequence. Besides, to align text prompts with image-level control, we design an attention masking mechanism.

to perform full attention in the temporal dimension, our work further introduces strategies like mixup to better exploit the rich priors inherent in video models.

# 3 Method

Given that video generative models' inherent **temporal full-attention** and **rich dynamics priors**, we argue they can be efficiently re-purposed for controllable image generation tasks. To successfully adapt smooth-transition-capable video generative models for handling abrupt and discontinuous image transitions, we propose multiple strategies, as shown in Figure 2. Specifically, in Section 3.1, we introduce our foundational model, HunyuanVideo-I2V, detailing its architecture and objective function; in Section 3.2, we present our mixup-based shot transition strategy that construct a shot transition video with condition and target images; in Section 3.3, we propose a new position embedding method that reduces the required number of transition frames; in Section 3.4, we describe an attention masking strategy to properly guide information interaction.

## 3.1 Preliminaries

Our method builds upon HunyuanVideo-I2V [20], which consists of three key components: (1) a causal 3DVAE that compresses videos in both spatial and temporal dimensions, (2) a text encoder built upon a Multimodal Large Language Model (MLLM), which processes not only textual information but also partial conditioning image features, (3) a transformer employing a unified full-attention mechanism to jointly process image and text signals.

The 3DVAE maps a video sequence $\mathbf{x} \in \mathbb{R}^{(4T+1) \times 3 \times 16H \times 16W}$ into a compact latent representation $\mathbf{y} \in \mathbb{R}^{(T+1) \times 16 \times 2H \times 2W}$, which is subsequently patchified and unfolded to yield visual tokens $\mathbf{z}_{visual}$ of length $(T + 1) \times H \times W$. Meanwhile, the textual tokens $\mathbf{z}_{textual}$ are obtained by processing target prompt $T_P$ and condition image $C_I$ through the MLLM. Then a concatenated sequence $\mathbf{z} = [\mathbf{z}_{visual}, \mathbf{z}_{textual}]$ is fed into the transformer, where a unified full-attention mechanism is applied to effectively fuse information across both modalities. To enhance the model's ability to capture positional relationships, 3D Rotary Position Embedding (RoPE) [41] is introduced in each transformer block. To achieve I2V generation, HunyuanVideo-I2V employs a token replacement technique, where the visual tokens of the first frame are replaced with the condition image tokens. In addition, CLIP-Large [37] text features and the diffusion timestep $t$ are adopted as global guidance signals and incorporated into the transformer. The objective function follows flow matching [28]:

$$\mathcal{L} = \|v_\theta(\mathbf{y}_t, t, C_I, T_P) - (\epsilon - \mathbf{y})\|^2, \tag{1}$$

where $\epsilon$ denotes Gaussian noise, $\mathbf{y}_t = (1 - t)\mathbf{y} + t\epsilon$, and $v$ and $\theta$ stand for the neural network and its corresponding parameters respectively.

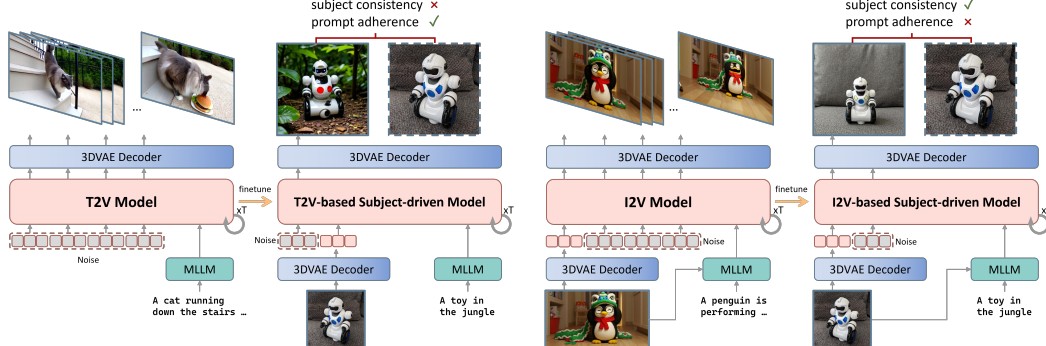

Figure 3: The inference process of T2V/I2V models and their finetuned subject-driven image generation models. By treating the condition and target images directly as a two-frame video and fine-tuning T2V/I2V models accordingly, the corresponding T2V/I2V baselines can be obtained.

## 3.2 Mixup-based Shot Transition

The simplest approach for controllable image generation using video generative models is to treat condition and target images as a two-frame video. During training, the condition image remains noiseless and excluded from loss calculation, while the target image is noise-corrupted and included in loss calculation. During inference, the condition image maintains noiseless to provide complete control signals. Empirical tests with HunyuanVideo-T2V/I2V on subject-driven generation task, as shown in Figure 3 and Table 3, demonstrate that neither model meets the requirements for subject-driven generation: the T2V model lacks subject consistency, while the I2V model over-preserves similarity to the condition image and exhibits poor prompt adherence. The observed results are expected because T2V model does not enforce consistency as strictly as I2V model, while the I2V model's strong inter-frame consistency preservation limits prompts' controllability.

To address these limitations, we draw inspiration from cinematic shot transitions by treating condition and target images as storyboard endpoints. Then, we fine-tune the I2V model to generate transition frames and target image according to condition image. This approach maintains consistency and enhances controllability through smooth visual transitions. Specifically, we observe that certain I2V models [20, 45] can naturally produce fade-in-fade-out transitions similar to those in PowerPoint presentations.Therefore, we propose constructing transition frames $F_\alpha$ with condition image $F_{\alpha=0}$ and target image $F_{\alpha=1}$ by interpolation, $F_\alpha = ((1-\beta) F_{\alpha=0}^\gamma + \beta F_{\alpha=1}^\gamma)^{1/\gamma}$, $\beta = \alpha^2 (3 - 2\alpha)$, where $\alpha \in [0, 1]$ and $\gamma$ is set to 2.2 ensure smooth inter-frame transitions. During training, we keep the condition image $F_{\alpha=0}$ noise-free and exclude it from loss calculation, while applying noise and including $F_{0<\alpha\leq1}$ in the loss calculation. The contribution weight of each latent frame in the loss is determined by its proportional content from the target image, yielding the final loss function:

$$\mathcal{L} = \frac{1}{K+1} \sum_{k=0}^{K} w(k) \|v_\theta(\mathbf{y}_t^k, t, C_I, C_P, T_P) - (\epsilon - \mathbf{y})\|^2,$$

$$\mathbf{y}_t^k = \begin{cases} (1-t) \cdot Encode(F_{0 \leq \alpha < 1}, k) + t\epsilon, & \text{if } k = 0, 1, \ldots, K-1, \\ (1-t) \cdot Encode(F_{\alpha=1}, -1) + t\epsilon, & \text{if } k = K, \end{cases} \tag{2}$$

$$w(k) = \frac{1}{4} \sum_{i=1}^{4} \left( \left( \frac{4k+i}{4K+1} \right)^2 \left( 3 - 2\frac{4k+i}{4K+1} \right) \right)^2,$$

where $Encode(\cdot, k)$ is the encoder of the 3DVAE, which maps $4T+1$ frames in pixel space to $T+1$ latent representations and returns the $(k+2)$-th latent representation, $C_P$ is the prompt of the condition image. We encode the target image separately to ensure the independence of the corresponding latent representation during inference. During inference, the condition image's latent representation is concatenated with $K+1$ Gaussian noise in latent space and perform progressive denoising while keeping the condition image's latent representation unchanged throughout the process, ultimately decoding the last frame of the denoised latent representations through the decoder $Decode(\cdot)$ of the 3DVAE to obtain the final generated result $\hat{F}_{\alpha=1} = Decode(\mathbf{y}^K)$.

### 3.3 Frame Skip Position Embedding

Achieving smooth shot transition often requires dozens or even hundreds of frames. Since we only aim to obtain the final frame, inserting so many transition frames between condition and target images would severely degrade the efficiency of both training and inference. In HunyuanVideo, the model incorporates both temporal and spatial information $(n, i, j)$ into tokens through RoPE, where $n = 0, 1, \cdots, T$ represents the latent frame index of the tokens in temporal dimension and $i = 0, 1, \cdots, H - 1$ and $j = 0, 1, \cdots, W - 1$ denote the height and width coordinates of the tokens in spatial dimensions, respectively. To achieve long-term effects with minimal latent frames, we enhance RoPE by incorporating skip intervals along the temporal dimension, called Frame Skip Position Embedding (FSPE), $(n', i', j') = (n \times \delta, i, j)$, where $\delta$ represents the skip interval. This approach constructs a long-term sparse representation of latent frames using minimal latent frames, significantly reducing computational overhead.

### 3.4 Attention Masking Strategy

Due to the absence of textual descriptions for shot transition videos, we jointly input the prompts from both condition and target image into the network on subject-driven generation task. This approach enables the model to acquire all textual information corresponding to the shot-transition videos. However, in this way, there are four distinct token sequences during full-attention computation, i.e., condition image tokens $C_I$, generated frame tokens $T_I$, target image prompt tokens $T_P$, and condition image prompt tokens $C_P$. To prevent unintended information blending across these token sequences, we design an attention masking strategy as illustrated in Figure 2. Specifically, our designed attention mask assigns a extremely negative value to similarity scores between incompatible token sequences (e.g., condition image tokens and target image prompt tokens) to effectively block unintended interactions while maintaining necessary information flows,

$$
M_{pq} = \begin{cases} -\infty, & \text{if } (p, q) \in (C_I \times T_I) \cup (T_I \times C_P) \cup (T_P \times C_I) \cup (T_P \times C_P) \cup (C_P \times T_I), \\ 0, & \text{otherwise.} \end{cases}
$$

(3)

Furthermore, during inference, we enhance the differentiation between $T_P$ and $C_P$ influences by augmenting the attention mask region corresponding to $(T_I \times T_P)$ with an offset of $\omega$ times its absolute mean value, where we set $\omega = 0.6$.

## 4 Experiments

### 4.1 Experimental Setup

**Tasks.** We extensively evaluate the effectiveness of our method across multiple tasks, including spatially-aligned generation, subject-driven generation and style tranferring. For spatially-aligned image generation, we specifically design five distinct sub-tasks: canny-to-image generation, depth-to-image generation, image colorization, image deblurring and image in/out-painting.

**Training.** For spatially-aligned image generation, we adopt a subset of the Text-to-Image-2M dataset [64] for training, consisting of around 160K samples, where the condition images are extracted from the corresponding ground-truth images. The models are trained with a batch size of 8 and gradient accumulation over 2 steps, resulting in an effective batch size of 16. We employ the AdamW optimizer and conduct training on 2 NVIDIA H800 GPUs (80GB memory each). For subject-driven image generation, we utilize the high-quality subset of the Subjects200K dataset [43], comprising approximately 110K image pairs for training. This model is trained using 4 NVIDIA H800 GPUs.

**Benchmarks.** For spatially-aligned generation, we employ the COCO2017 validation dataset [26] comprising 5,000 images resized to $512 \times 512$ resolution as the test set, where the corresponding prompts are randomly selected from multiple candidate captions associated with each image. For subject-driven generation, we evaluate our method on DreamBench [40] by generating images for 25 text prompts per subject, using one reference image for each of the 30 subjects in the benchmark.

**Metrics.** For spatially-aligned generation, we evaluate methods in terms of controllability and generation quality. Controllability is assessed by the similarity of the extracted condition images from generated and ground-truth image. Specifically, we employ the F1 score for canny-to-image

Table 1: Quantitative results on COCO2017 validation set. The best results are in **bold**.

| Condition | Model | Method | Controllability F1↑/MSE↓ | General Quality FID↓ | SSIM↑ |
|-----------|-------|--------|:---:|:---:|:---:|
| Canny | SD1.5 [38] | ControlNet [58] | 0.34 | 18.74 | 0.35 |
| | | T2I-Adapter [31] | 0.22 | 20.06 | 0.35 |
| | | Uni-ControlNet [62] | 0.20 | 17.38 | – |
| | FLUX.1 [23] | ControlNet | 0.21 | 98.68 | 0.25 |
| | | OminiControl [43] | 0.38 | 20.63 | **0.40** |
| | | EasyControl [61] | 0.31 | **16.07** | – |
| | HunyuanVideo-I2V [20] | `DRA-Ctrl` | **0.42** | 19.44 | 0.38 |
| Depth | SD1.5 | ControlNet | 923 | 23.02 | 0.34 |
| | | T2I-Adapter | 1560 | 24.72 | 0.27 |
| | | Uni-ControlNet | 1685 | 21.79 | – |
| | FLUX.1 | ControlNet | 2958 | 62.20 | 0.26 |
| | | OminiControl | 903 | 27.26 | **0.39** |
| | | EasyControl | 1092 | **20.39** | – |
| | HunyuanVideo-I2V | `DRA-Ctrl` | **76** | 20.83 | 0.33 |
| Deblur | FLUX.1 | ControlNet | 572 | 30.38 | 0.74 |
| | | OminiControl | 132 | 11.49 | **0.87** |
| | HunyuanVideo-I2V | `DRA-Ctrl` | **11** | **9.08** | 0.64 |
| Colorization | FLUX.1 | ControlNet | 351 | 16.27 | 0.64 |
| | | OminiControl | **24** | 10.23 | 0.73 |
| | HunyuanVideo-I2V | `DRA-Ctrl` | 30 | **8.39** | **0.85** |
| Mask | SD1.5 | ControlNet | 7588 | 13.14 | 0.40 |
| | FLUX.1 | OminiControl | 6248 | 15.66 | 0.48 |
| | HunyuanVideo-I2V | `DRA-Ctrl` | **16** | **9.87** | **0.59** |

task and use Mean Squared Error (MSE) for other tasks. Generation quality is quantified using Fréchet Inception Distance (FID) [13] and Structural Similarity Index Measure (SSIM) [48] between generated and ground-truth images. For subject-driven generation, we evaluate methods by standard automatic metrics and a Vison-Language (VL) Model. We measure subject consistency by DINO and CLIP-I scores, which compute the cosine similarity between the condition image and the generated image in DINO [3] and CLIP [37] embedding spaces. Prompt adherence is quantified by the cosine similarity between the CLIP embeddings of the prompt and the generated image, referred to as CLIP-T score. However, these metrics have inherent limitations: DINO and CLIP-I measure global image similarity rather than directly evaluating subject consistency, while CLIP-T struggles with fine-grained semantic alignment and other challenges [37]. To address this, we propose VL score, a novel metric based on QWen2.5-VL [1], which evaluates generated images for subject consistency and prompt adherence via tailored prompts. The VL model outputs discrete scores (0-4) per dimension, with the final score computed as their average.

## 4.2 Spatially-aligned Image Generation Results

To validate `DRA-Ctrl`'s effectiveness for spatially-aligned generation tasks, we conduct comprehensive comparisons with multiple competitive approaches. As shown in Figure 4b, our method demonstrates superior performance in several aspects: compared to OminiControl, our approach generates more realistic traffic light images for canny-to-image; produces images with more vivid details for depth-to-image; achieves richer color variations in the blue-boxed regions for colorization; better preserves original image details in red-boxed areas for deblurring; and creates more authentic results for inpainting. These qualitative comparisons consistently highlight our method's advantages in maintaining spatial alignment while generating high-quality images across diverse generation scenarios. Quantitative results presented in Table 1 further demonstrate the superiority of `DRA-Ctrl`. `DRA-Ctrl` achieves significant advantages in controllability, attaining the best results across all tasks except colorization, while maintaining highly competitive performance in general quality.

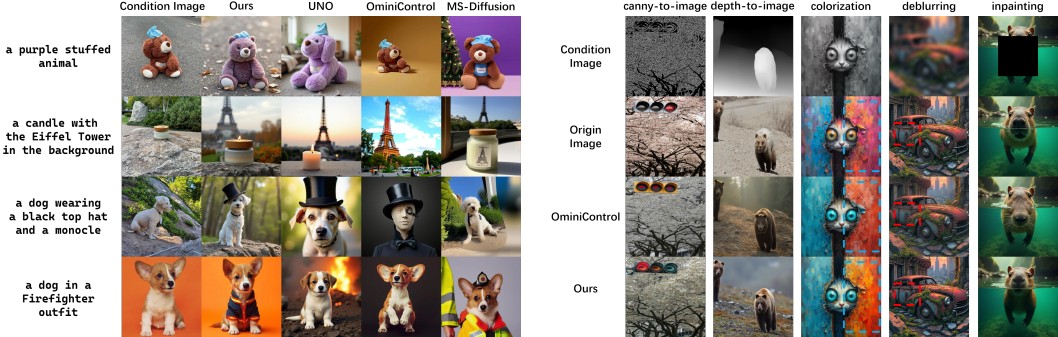

(a) Subject-driven image generation.  (b) Spatially-aligned image generation.

Figure 4: Qualitative results comparing different methods.

Table 2: Quantitative results on DreamBench. The **best** and **second best** values of each metric are highlighted.

| Method | VL Score↑ | DINO↑ | CLIP-I↑ | CLIP-T↑ |
|---|---|---|---|---|
| Oracle | – | 0.774 | 0.885 | – |
| Textual Inversion [11] | – | 0.569 | 0.780 | 0.255 |
| DreamBooth [40] | – | 0.668 | 0.803 | 0.305 |
| BLIP-Diffusion [24] | – | 0.670 | 0.805 | 0.302 |
| ELITE [49] | – | 0.647 | 0.772 | 0.296 |
| Re-Imagen [5] | – | 0.600 | 0.740 | 0.270 |
| BootPIG [35] | – | 0.674 | 0.797 | 0.311 |
| SSR-Encoder [60] | – | 0.612 | 0.821 | 0.308 |
| OmniGen [51] | – | 0.693 | 0.801 | 0.315 |
| OmniControl [43] | 2.21 | 0.559 | 0.765 | 0.310 |
| FLUX.1 IP-Adapter [54] | – | 0.582 | 0.820 | 0.288 |
| MS-Diffusion [46] | 1.94 | 0.655 | 0.782 | 0.307 |
| UniReal [7] | – | 0.702 | 0.806 | 0.326 |
| UNO [50] | 2.43 | 0.657 | 0.786 | 0.315 |
| DRA-Ctrl | 2.56 | 0.722 | 0.825 | 0.302 |

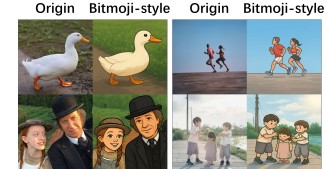

Figure 5: Qualitative results of `DRA-Ctrl` on style transferring.

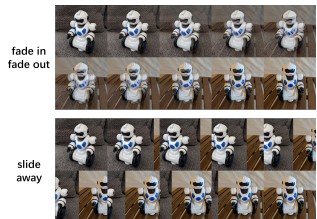

Figure 6: Different mixup-based shot transition types.

## 4.3 Subject-driven Image Generation Results

To validate the effectiveness of `DRA-Ctrl` for subject-driven generation, we conduct comprehensive comparisons with multiple state-of-the-art approaches. Qualitative results are presented in Figure 4a, where our method demonstrates superior subject consistency. As shown in the third row, our approach generates a dog that even preserves details like the neck tag, while competing methods exhibit inconsistent breeds or fail to generate the subject altogether. The quantitative results are presented in Table 2, where we compare various tuning-based and tuning-free approaches. Under all comparison methods, our approach achieves the highest VL Score (2.56), DINO (0.722), and CLIP-I (0.825), along with a competitive CLIP-T score of 0.302.

## 4.4 Style Transfer

We employ GPT-4o to generate 100 original-to-Bitmoji-style image pairs, which are subsequently used to fine-tune our subject-driven model for achieving style transfer effects. The results are demonstrated in Figure 5, where our model successfully captures the distinctive aesthetic characteristics of Bitmoji-style animation while preserving the original content's structural integrity.

## 4.5 Ablation Studies

To validate the effectiveness of our proposed strategies, we conduct comprehensive ablation studies on our method from multiple perspectives, including comparisons with T2V/I2V baselines, analysis of different shot transition types, ablation on the number of transition frames, and module ablation.

Table 3: Comparison with baselines.

| | VL↑ | DINO↑ | CLIP-I↑ | CLIP-T↑ |
|---|---|---|---|---|
| Oracle | – | 0.774 | 0.885 | – |
| T2V baseline | 2.01 | 0.658 | 0.787 | **0.306** |
| I2V baseline | 2.34 | **0.803** | **0.874** | 0.291 |
| DRA-Ctrl | **2.44** | 0.715 | 0.821 | 0.298 |

Table 4: Ablation on transition types.

| | VL↑ | DINO↑ | CLIP-I↑ | CLIP-T↑ |
|---|---|---|---|---|
| slide away | 2.19 | 0.708 | 0.822 | 0.292 |
| fade in fade out | **2.42** | **0.742** | **0.834** | **0.295** |

Table 5: Ablation on frame numbers.

| number of transition frames | VL↑ | DINO↑ | CLIP-I↑ | CLIP-T↑ |
|---|---|---|---|---|
| 4 | 2.19 | 0.692 | 0.820 | 0.292 |
| 8 | **2.42** | **0.742** | **0.834** | **0.295** |
| 12 | 2.09 | 0.715 | 0.826 | 0.283 |

Table 6: Ablation on modules in DRA-Ctrl.

| | VL↑ | DINO↑ | CLIP-I↑ | CLIP-T↑ |
|---|---|---|---|---|
| Oracle | – | 0.774 | 0.885 | – |
| w/o loss reweighting | 2.32 | 0.744 | 0.839 | 0.292 |
| w/o FSPE | 2.28 | 0.777 | 0.853 | 0.287 |
| w/o mixup strategy | 2.38 | **0.900** | **0.918** | 0.271 |
| w/o attention masking | 2.41 | 0.777 | 0.856 | 0.292 |
| full version | **2.42** | 0.742 | 0.834 | **0.295** |

Table 7: Generation efficiency analysis.

| | latent frames | VL↑ | DINO↑ | CLIP-I↑ | CLIP-T↑ | Time/s↓ |
|---|---|---|---|---|---|---|
| Oracle | – | – | 0.774 | 0.885 | – | – |
| I2V baseline | 2 | 2.34 | **0.803** | **0.874** | 0.291 | **10.8** |
| DRA-Ctrl | 4 | **2.44** | 0.715 | 0.821 | **0.298** | 24.0 |
| I2V | 37 | 1.09 | 0.698 | 0.810 | 0.257 | 251 |

**Comparison between T2V/I2V baselines.** Quantitative results 3 on DreamBench align with Figure 3 and Section 3.2. The T2V baseline, whose base model is unable to accept images as control signals, achieves a high CLIP-T score but suffers from low DINO and CLIP-I scores. The I2V baseline produces condition image-like outputs, with the DINO score even surpassing the result measured on real images, but suffers from low prompt adherence. Under identical experimental configurations, DRA-Ctrl achieves a balanced performance, with DINO, CLIP-I and CLIP-T positioned between the two baselines and the highest VL Score, exhibiting superior performance.

**Different mixup-based shot transition types.** In addition to the fade-in-fade-out approach for constructing transition frames, we also experimented with slide-away transitions, with examples illustrated in Figure 6. Quantitative results in Table 4 demonstrate that the fade-in-fade-out mixup strategy outperforms slide-away across all three metrics. This observation aligns with our findings that video models tend to exhibit stronger priors for fade-in-fade-out shot transitions, while showing weaker priors for more complex transition types.

**Number of transition frames.** We investigate the impact of varying numbers of transition frames on experimental results, as shown in Table 5. Both insufficient and excessive transition frames harm performance. This phenomenon may stem from two factors: too few frames create excessively large inter-frame variations that increase learning difficulty, while too many frames introduce unnecessary computational overhead and slower convergence under the same training budget.

**Module ablation.** We conduct ablation studies on our proposed modules, including loss reweighting, FSPE, mixup strategy, and attention masking, with experimental results summarized in Table 6. Since our method employs an I2V model as the base architecture, all proposed modules aim to address its inherent limitations of excessive similarity to the condition image and poor prompt adherence. The results demonstrate that FSPE, mixup strategy, and attention masking significantly mitigate these issues, while loss reweighting primarily accelerates model convergence.

## 4.6 Generation Effiency Analysis

To analyze DRA-Ctrl's generation efficiency, we compare against the I2V baseline and the I2V model on DreamBench, assessing generation quality and efficiency. The I2V model generates videos from prompts and condition images, using the final frames as outputs. With $\delta = 12$ in FSPE (corresponding to 48 pixel-space frames), we set the I2V model to produce 145-frame videos. Table 7 results show our method achieves 90.4% faster generation than the I2V model with the highest VL score.

# 5 Conclusion

Leveraging the rich high-dimensional information priors inherent in video models, we propose to repurpose them for low-dimensional controllable image generation, demonstrating advantages akin to a "*Dimensionality-Reduction Attack*" effect compared to conventional image generation models. Specifically, to bridge the gap between video models' native capability for modeling continuous smooth transitions and the requirement for discrete abrupt changes in controllable image generation, we introduce a novel mixup-based transition strategy that constructs smooth transition between condition image and target image. Moreover, we redesign the attention masking mechanism that precisely aligns text prompts with image-level control signals. Our work establishes a new paradigm for activating high-dimensional video models to solve low-dimensional image generation tasks, while paves the way for future development of unified generative models across visual modalities.

**Limitations.** Our method employs a video model not optimized for image generation, resulting in slightly inferior performance on image quality metrics (FID, SSIM) compared to image-specific approaches. Besides, since HunyuanVideo-I2V primarily uses LLaVA [8] for prompt understanding, our CLIP-T scores are marginally lower than competing methods. Additionally, the requirement for transitional frames leads to reduced generation efficiency.

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

# A  More Experimental Details

In this section, we provide additional experimental details, including the configurations of LoRA and other hyperparameters. For different tasks, we employ distinct settings: Section A.1 describes the spatially-aligned image generation tasks, Section A.2 covers the subject-driven image generation task, and Section A.3 presents the experimental details for style transfer.

`DRA-Ctrl` employs LoRA [14] to fine-tune the base model with a rank of 16. Since our method needs to simultaneously process noiseless condition image token sequences and noisy generated image token sequences, we set the LoRA scale to 0 when handling the generated image token sequences to distinguish between them. Additionally, we set $\delta$ to 12 in the Frame Skip Position Embedding (FSPE). This configuration enables 4 frames in the latent space to effectively emulate 37 frames, corresponding to $1 + 36 \times 4 = 145$ frames in pixel space — approximately equivalent to a 5-second short video at 30 frames per second (fps), which sufficiently achieves the shot transition effect.

## A.1  Spatially-aligned Image Generation

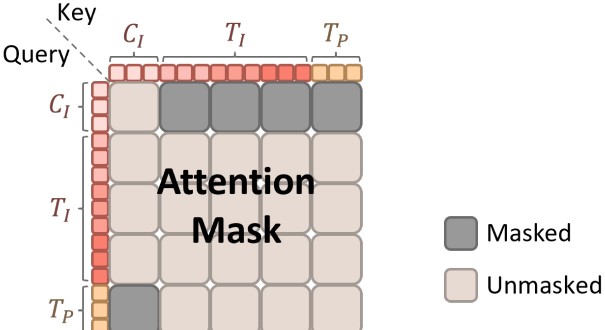

Figure 7: Attention masking strategy in spatially-aligned tasks.

In spatially-aligned image generation tasks, the condition image is directly extracted from the ground-truth image without a corresponding prompt. Therefore, we do not employ the condition image prompt $C_P$ in our experiments, but we still utilize the attention masking strategy, with the corresponding attention mask illustrated in Figure 7. Besides, we train the model for 6,000 steps. In depth-to-image and depth prediction tasks, the depth image is extracted from the ground-truth image using Depth Anything [52]. For the depth prediction task, we prepend "[depth] " to the prompt to guide the model to generate depth maps rather than regular images. In the deblurring task, we apply Gaussian blur to the images with a randomly selected integer blur radius between 1 and 10 during training. For the in/out-painting task, we randomly select a rectangular region in the image during training, then mask either the selected region (with 0.5 probability) or the area outside it (with 0.5 probability) to create the condition image. In the super-resolution task, the condition image is obtained by downsampling the original image by a factor of 4.

## A.2  Subject-driven Image Generation

For the subject-driven image generation task, we train the model for 9,000 steps. During inference, while employing attention masking, the simultaneous presence of both target image prompts $T_P$ and condition image prompts $C_P$ may still cause information blending. To address this, we strengthen the interaction between target image tokens $T_I$ and $T_P$ while suppressing $C_P$'s influence on the generated output. Specifically, within the $(T_I \times T_P)$ attention mask region, we augment the attention weights by adding $0.6 \times \mu$ (where $\mu$ denotes the mean absolute value of the original weights). The modified attention computation for this region is formulated as:

$$\text{Attention}(Z) = \text{softmax}\left( \frac{Q_Z K_Z^\top}{\sqrt{d}} + 0.6 \times \text{mean}\left( \left| \frac{Q_Z K_Z^\top}{\sqrt{d}} \right| \right) \right) V_Z. \tag{4}$$

## A.3  Style Transfer

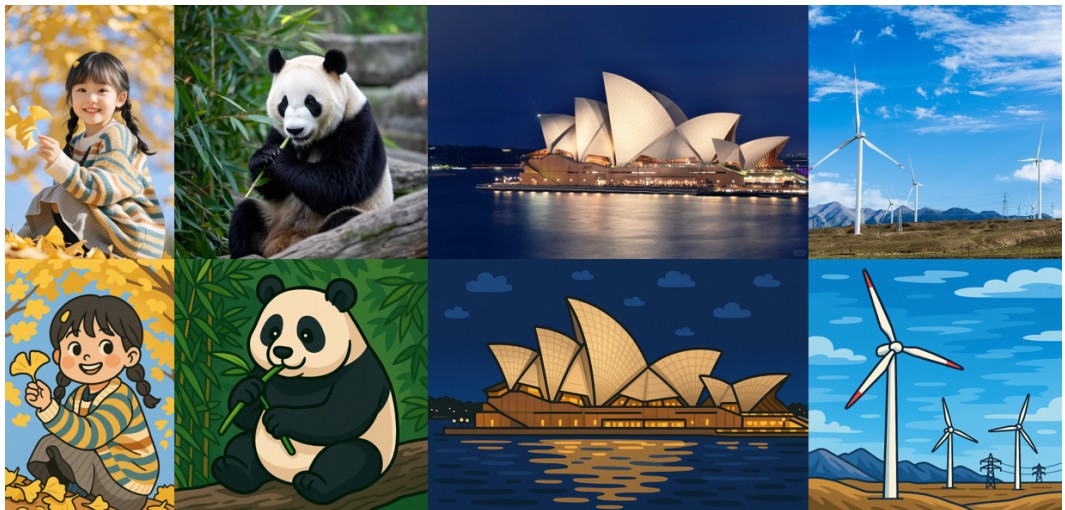

Figure 8: Bitmoji-style example images in our dataset.

```
[USER PROMPT]:
将上传的图像分别转换为 bitmoji 风格，尺寸大小为{}:{}，输出清晰的图像。
```

Figure 9: The prompt format used for generating Bitmoji-style images with GPT-4o.

We collected 100 diverse images containing subjects such as humans, animals and buildings from the web. Using carefully designed prompts, we guided ChatGPT-4o to generate corresponding Bitmoji-style images, which formed our training set. The subject-driven image generation model is fine-tuned for 2,600 steps with a batch size of 8 on an NVIDIA H800 GPU to obtain the final model. Example images from our dataset are shown in Figure 8, and the prompt format we employed is shown in Figure 9, where the image dimensions are determined by their original resolutions.

## B  More Details about the VL Score

Current evaluation metrics for subject-driven image generation primarily employ DINO and CLIP-I to assess subject consistency, and CLIP-T for prompt adherence. However, two critical limitations exist: first, there lacks a comprehensive metric to directly evaluate subject-driven generation quality; second, these existing metrics exhibit notable shortcomings — both DINO and CLIP-I are significantly influenced by background interference, while CLIP-T struggles with fine-grained semantic alignment.

To address these issues, we propose leveraging an advanced Vision-Language (VL) model, such as QWen2.5-VL [1], as an evaluator to produce a holistic metric. Our approach consists of three steps: First, we provide the VL model with a prompt instructing it to score (prompt, reference image, generated image) triplets based on multiple fine-grained criteria for both subject consistency and prompt adherence. Next, we have the model summarize its task to confirm proper understanding. Finally, we input each triplet and collect the model's scores. Since both metrics are discrete scores ranging from 0 to 4, we average them to derive a comprehensive metric termed the VL Score. An example input-output demonstration of the VL model is shown in Figure 10.

## C  More Visualization

This section presents additional qualitative experimental results across all tasks, including transition frames generated by our model. The spatially-aligned image generation results are detailed in Section C.1, while the subject-driven image generation outcomes are presented in Section C.2, and the style transfer performance is analyzed in Section C.3. Unless otherwise specified, all image generation in this paper uses 50 sampling steps by default, including both qualitative results and quantitative evaluations, and generated images maintain a consistent resolution of $512 \times 512$ pixels.

```
[USER PROMPT]:
[Task Instruction] As a professional evaluator, your role is to assess the semantic consistency between a
↪  reference image and an generated image based on a given text prompt. Focus on two aspects:
1. Subject Consistency: Compare key visual attributes (Shape: main body outline, structure, and
↪  proportions. Color: saturation, hue, brightness, and distribution. Texture: realism, clarity, and
↪  aesthetic appeal. Facial: facial features, only apply when subject is a person or animal) between the
↪  reference and generated image.
2. Prompt Adherence: Verify if the generated image accurately reflects all elements described in the text
↪  prompt (Relevance: whether core elements in text are shown. Accuracy: objects, settings, actions, or
↪  characteristics Completeness: whether the generated image omits significant details. Context:
↪  portrayal of the appropriate environment, interactions, and background elements).
You need to provide your scores Subject Consistency and Prompt Adherence based on the following criteria.
[Subject Consistency Scoring Criteria] Very Poor (0): No resemblance. The generated image's subject has
↪  no relation to the reference.
Poor (1): Minimal resemblance. The subject falls within the same broad category but differs
↪  significantly.
Fair (2): Moderate resemblance. The subject shows likeness to the reference with notable variances.
Good (3): Strong resemblance. The subject closely matches the reference with only minor discrepancies.
Excellent (4): Near-identical. The subject of the generated image is virtually indistinguishable from the
↪  reference.
[Prompt Adherence Scoring Criteria] Very Poor (0): No resemblance. The generated image's subject has no
↪  relation to the reference.
Poor (1): Minimal resemblance. The subject falls within the same broad category but differs
↪  significantly.
Fair (2): Moderate resemblance. The subject shows likeness to the reference with notable variances.
Good (3): Strong resemblance. The subject closely matches the reference with only minor discrepancies.
Excellent (4): Near-identical. The subject of the generated image is virtually indistinguishable from the
↪  reference.
[Input Format] Every time you will receive two images and a prompt, the first image is a reference image,
↪  and the second image is the generated image, and the text is the prompt describing the generated
↪  image.
[Output Format] return the scores (0-4) in JSON format, strictly following this example:
↪  {\"subject_consistency\": 3, \"prompt_adherence\": 4}.

[VL MODEL OUTPUT]:
Sure, I understand the task. Please provide me with the reference image, the generated image, and the
↪  text prompt so that I can evaluate them according to the provided criteria.

[USER PROMPT]:
Did you understand the task above? Please summarize the tasks you need to do and show how you will
↪  execute the detailed plan for the task.

[VL MODEL OUTPUT]:
Yes, I understood the task clearly. too long, omitted

[USER PROMPT]:
{the first image} {the second image} {prompt}

[VL MODEL OUTPUT]:
```json\n{"subject_consistency": 3, "prompt_adherence": 2}\n```

More...
```

Figure 10: An example of VL Score evaluation process.

## C.1 Spatially-aligned Image Generation Results

Our method performs image-to-video generation conditioned on input images, where the state of these condition images significantly impacts the output quality. We found that directly using canny edges, depth maps with black representing maximum depth, or black masks in in/out-painting tasks often resulted in unnaturally dark generated images. To address this, we implemented a color normalization scheme that remaps the darkest values $(0, 0, 0)$ to medium-gray $(128, 128, 128)$ while linearly scaling all other color values proportionally, preventing extreme darkening.

### C.1.1 Canny-to-image

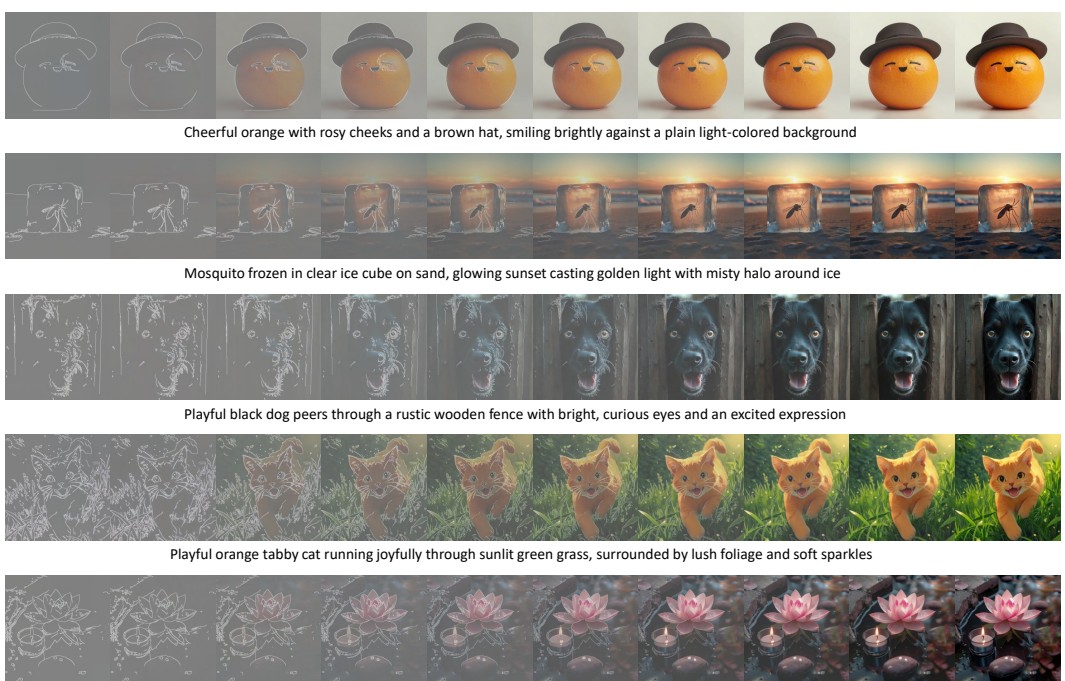

Cheerful orange with rosy cheeks and a brown hat, smiling brightly against a plain light-colored background

Mosquito frozen in clear ice cube on sand, glowing sunset casting golden light with misty halo around ice

Playful black dog peers through a rustic wooden fence with bright, curious eyes and an excited expression

Playful orange tabby cat running joyfully through sunlit green grass, surrounded by lush foliage and soft sparkles

Serene lotus with pink petals floats near a glowing candle on dark reflective water, surrounded by stones

Figure 11: More canny-to-image generation results.

### C.1.2 Colorization

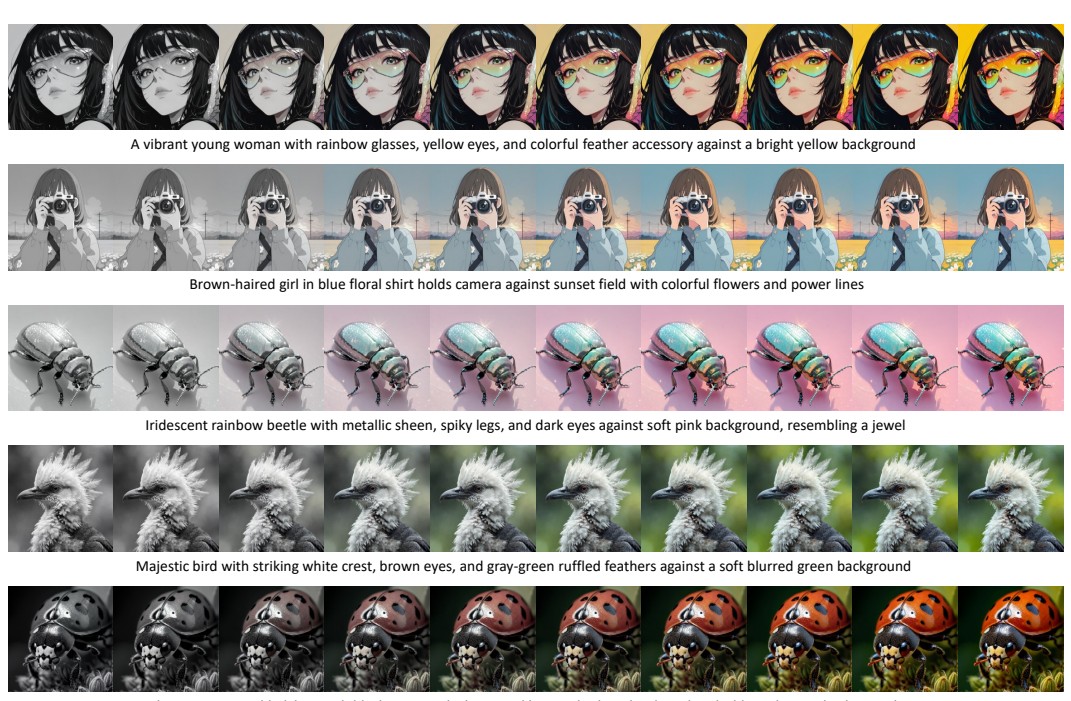

A vibrant young woman with rainbow glasses, yellow eyes, and colorful feather accessory against a bright yellow background

Brown-haired girl in blue floral shirt holds camera against sunset field with colorful flowers and power lines

Iridescent rainbow beetle with metallic sheen, spiky legs, and dark eyes against soft pink background, resembling a jewel

Majestic bird with striking white crest, brown eyes, and gray-green ruffled feathers against a soft blurred green background

Vibrant orange-red ladybug with black spots and white speckles perched on dried seed pods, blurred green background

Figure 12: More colorization generation results.

## C.1.3 Deblurring

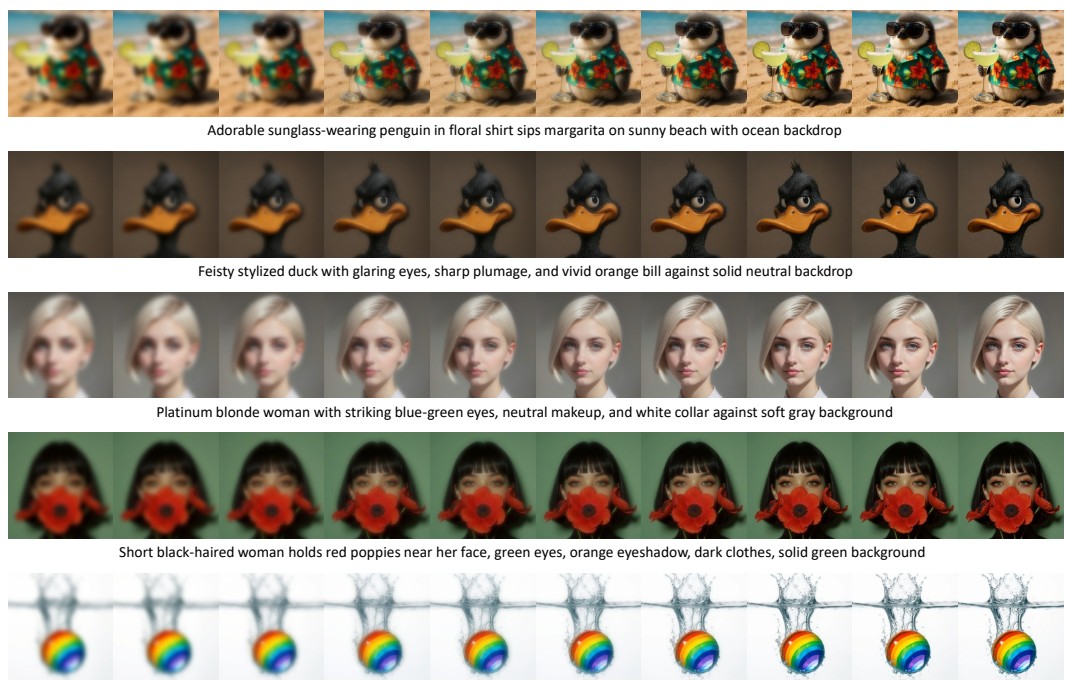

Adorable sunglass-wearing penguin in floral shirt sips margarita on sunny beach with ocean backdrop

Feisty stylized duck with glaring eyes, sharp plumage, and vivid orange bill against solid neutral backdrop

Platinum blonde woman with striking blue-green eyes, neutral makeup, and white collar against soft gray background

Short black-haired woman holds red poppies near her face, green eyes, orange eyeshadow, dark clothes, solid green background

Vibrant rainbow ball creates dramatic splash in clear water, bubbles swirling against crisp white background

Figure 13: More deblurring generation results.

## C.1.4 Depth-to-image

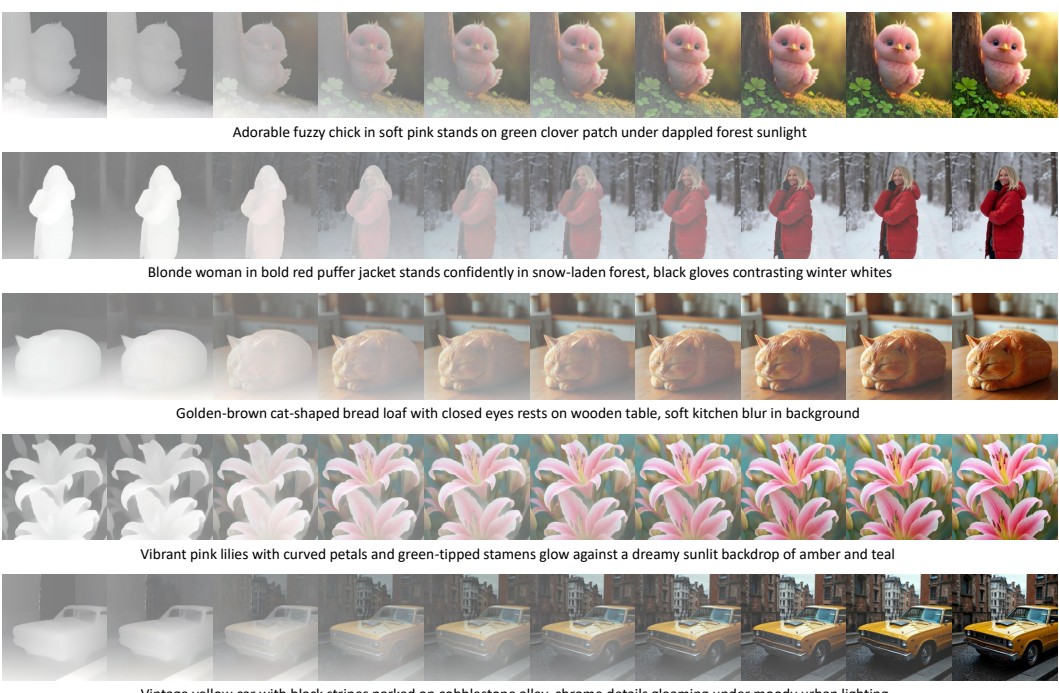

Adorable fuzzy chick in soft pink stands on green clover patch under dappled forest sunlight

Blonde woman in bold red puffer jacket stands confidently in snow-laden forest, black gloves contrasting winter whites

Golden-brown cat-shaped bread loaf with closed eyes rests on wooden table, soft kitchen blur in background

Vibrant pink lilies with curved petals and green-tipped stamens glow against a dreamy sunlit backdrop of amber and teal

Vintage yellow car with black stripes parked on cobblestone alley, chrome details gleaming under moody urban lighting

Figure 14: More depth-to-image generation results.

### C.1.5 Depth Prediction

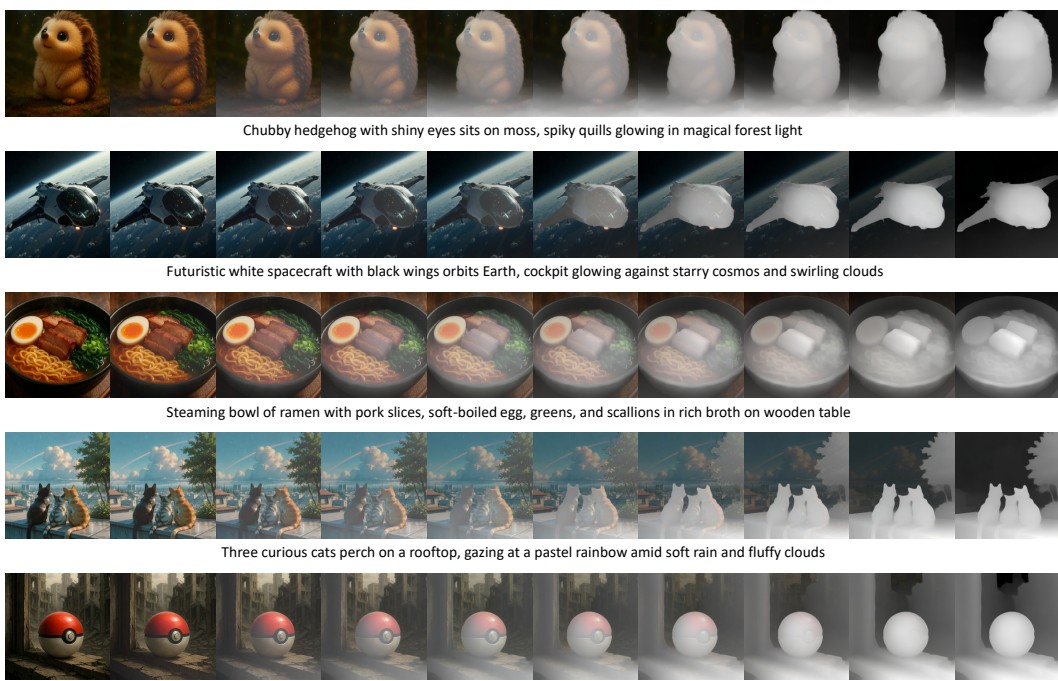

Chubby hedgehog with shiny eyes sits on moss, spiky quills glowing in magical forest light

Futuristic white spacecraft with black wings orbits Earth, cockpit glowing against starry cosmos and swirling clouds

Steaming bowl of ramen with pork slices, soft-boiled egg, greens, and scallions in rich broth on wooden table

Three curious cats perch on a rooftop, gazing at a pastel rainbow amid soft rain and fluffy clouds

Vibrant red Pokéball rests on cracked ledge amid ruined city's gray rubble and eerie silence

Figure 15: More image-to-depth generation results.

### C.1.6 In/out-painting

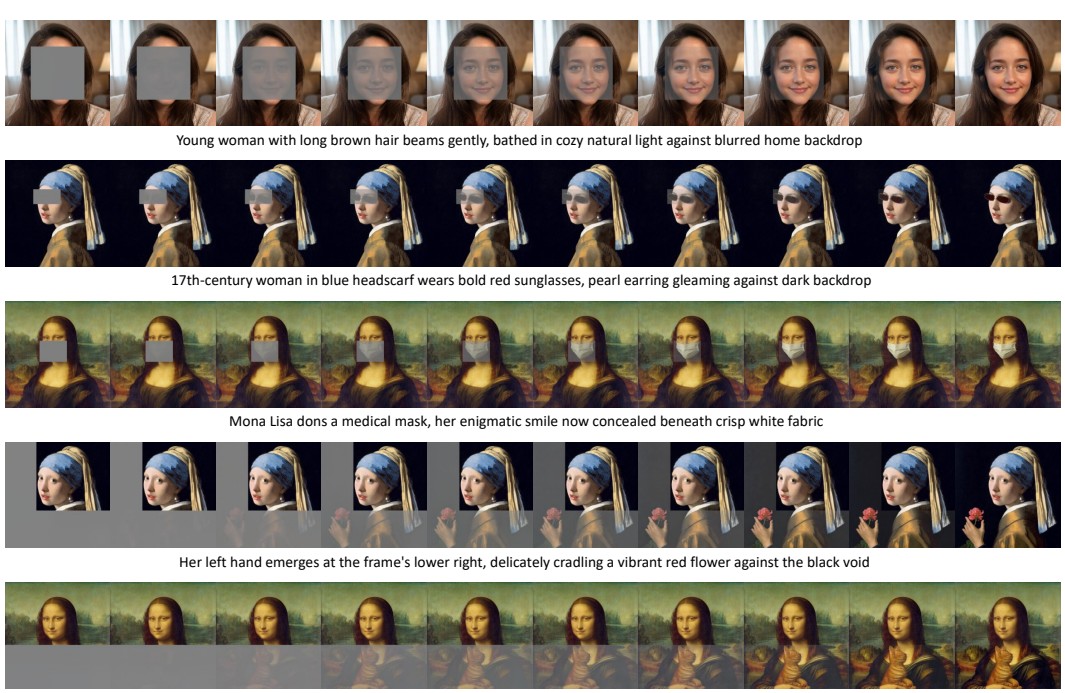

Young woman with long brown hair beams gently, bathed in cozy natural light against blurred home backdrop

17th-century woman in blue headscarf wears bold red sunglasses, pearl earring gleaming against dark backdrop

Mona Lisa dons a medical mask, her enigmatic smile now concealed beneath crisp white fabric

Her left hand emerges at the frame's lower right, delicately cradling a vibrant red flower against the black void

In the lower half of the image, Mona Lisa is holding an adorable orange kitten

Figure 16: More in/out-painting generation results.

### C.1.7 Super-resolution

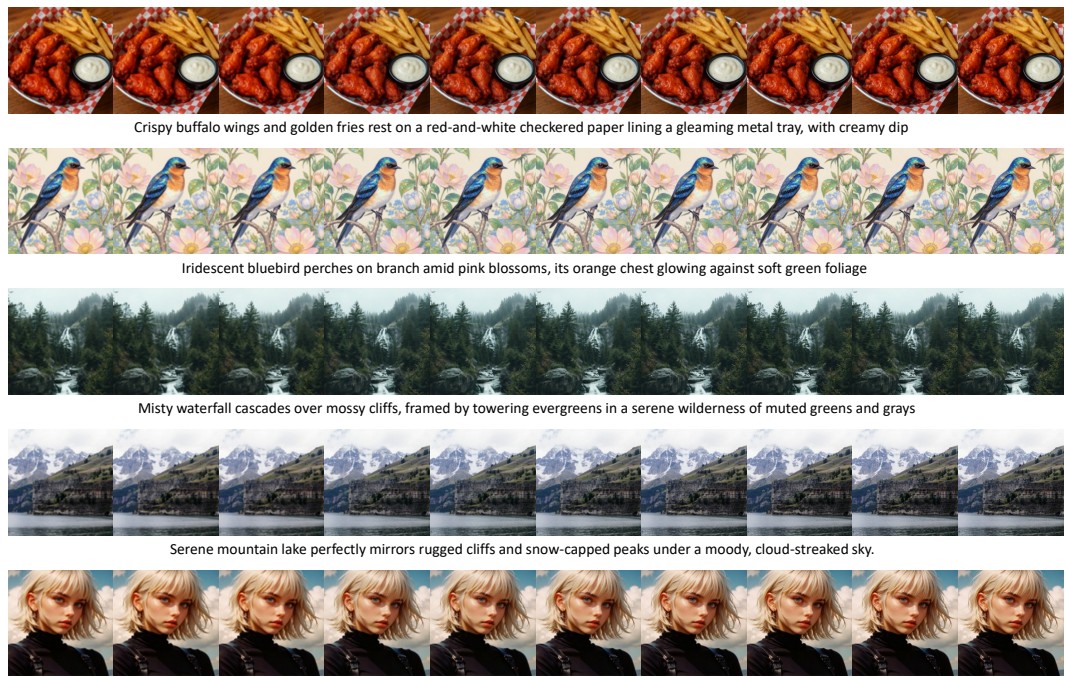

Crispy buffalo wings and golden fries rest on a red-and-white checkered paper lining a gleaming metal tray, with creamy dip

Iridescent bluebird perches on branch amid pink blossoms, its orange chest glowing against soft green foliage

Misty waterfall cascades over mossy cliffs, framed by towering evergreens in a serene wilderness of muted greens and grays

Serene mountain lake perfectly mirrors rugged cliffs and snow-capped peaks under a moody, cloud-streaked sky.

Stylish blonde in black turtleneck and suspenders gazes confidently against a bright blue sky with fluffy clouds

Figure 17: More super-resolution generation results.

### C.2 Subject-driven Image Generation Results

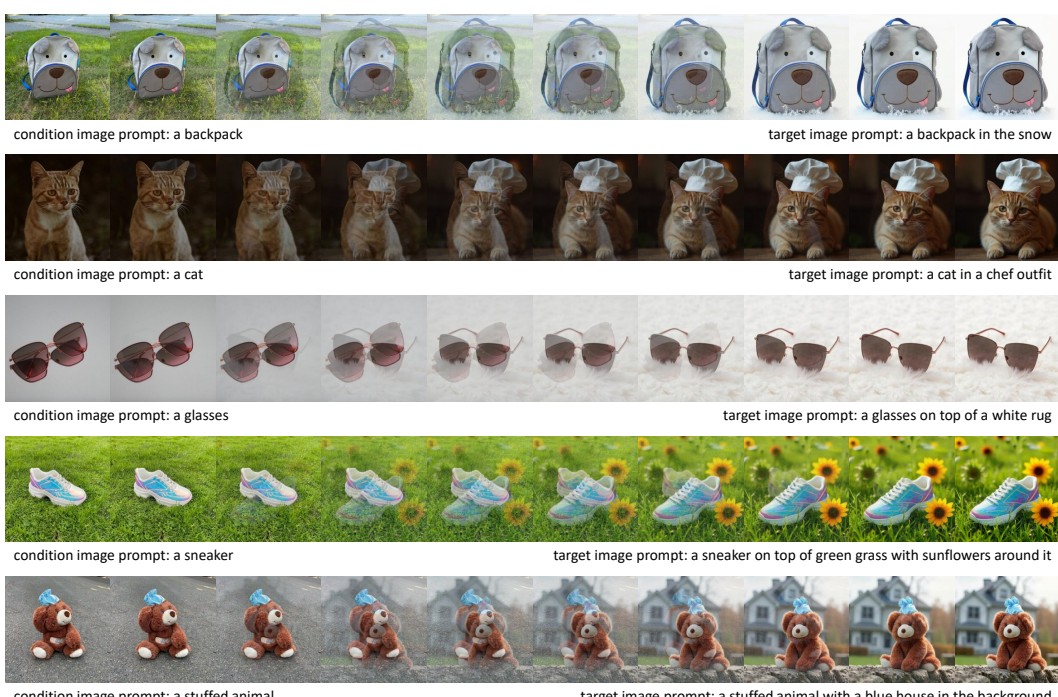

condition image prompt: a backpack                    target image prompt: a backpack in the snow

condition image prompt: a cat                    target image prompt: a cat in a chef outfit

condition image prompt: a glasses                    target image prompt: a glasses on top of a white rug

condition image prompt: a sneaker                    target image prompt: a sneaker on top of green grass with sunflowers around it

condition image prompt: a stuffed animal                    target image prompt: a stuffed animal with a blue house in the background

Figure 18: More subject-driven generation results on DreamBench.

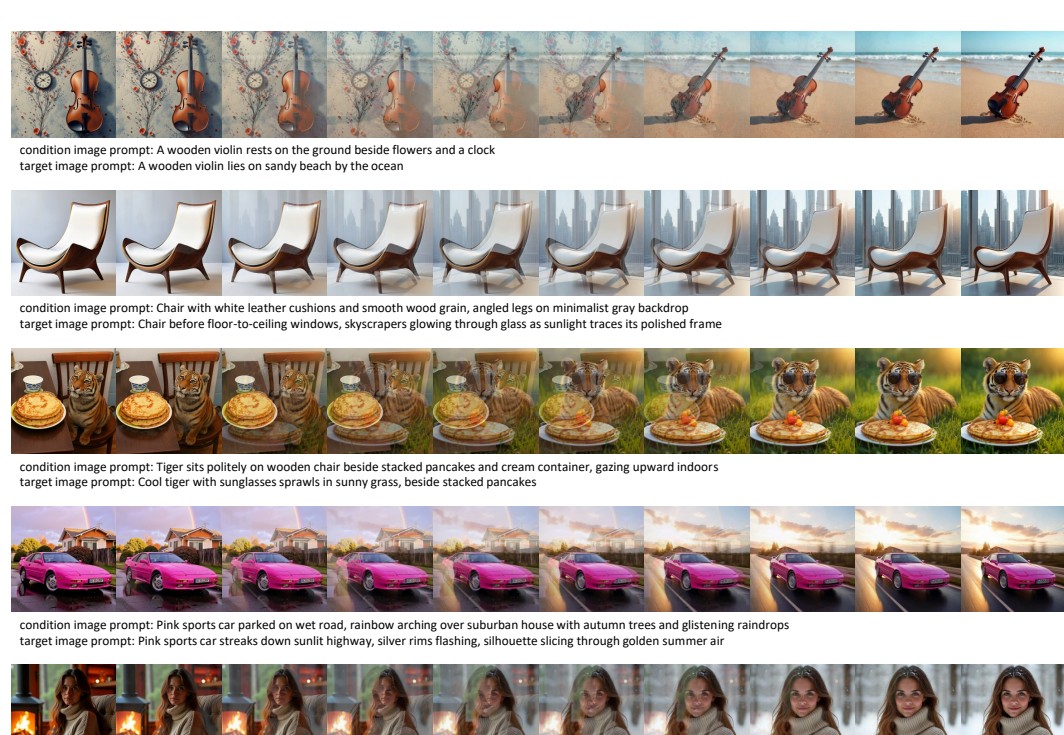

condition image prompt: A wooden violin rests on the ground beside flowers and a clock
target image prompt: A wooden violin lies on sandy beach by the ocean

condition image prompt: Chair with white leather cushions and smooth wood grain, angled legs on minimalist gray backdrop
target image prompt: Chair before floor-to-ceiling windows, skyscrapers glowing through glass as sunlight traces its polished frame

condition image prompt: Tiger sits politely on wooden chair beside stacked pancakes and cream container, gazing upward indoors
target image prompt: Cool tiger with sunglasses sprawls in sunny grass, beside stacked pancakes

condition image prompt: Pink sports car parked on wet road, rainbow arching over suburban house with autumn trees and glistening raindrops
target image prompt: Pink sports car streaks down sunlit highway, silver rims flashing, silhouette slicing through golden summer air

condition image prompt: Woman in cream knit sweater sits calmly by a crackling fireplace, surrounded by warm candlelight and rustic wooden shelves
target image prompt: The woman stands in a snowy forest, captured in a half-portrait

Figure 19: More subject-driven generation results.

Interestingly, we discover that during subject-driven image generation, `DRA-Ctrl` can occasionally control two subjects in the condition image simultaneously. As shown in the third row of Figure 19, our method successfully makes the tiger wear sunglasses while placing the stacked pancakes on the grass.

## C.3 Style Transfer

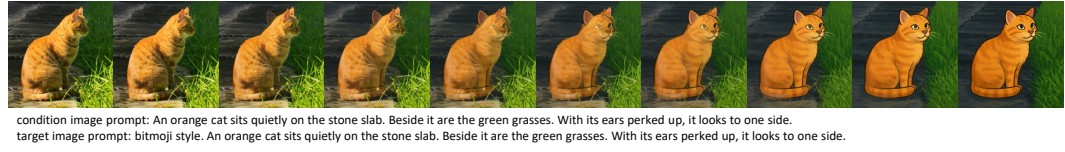

condition image prompt: An orange cat sits quietly on the stone slab. Beside it are the green grasses. With its ears perked up, it looks to one side.
target image prompt: bitmoji style. An orange cat sits quietly on the stone slab. Beside it are the green grasses. With its ears perked up, it looks to one side.



condition image prompt: The back view of an old couple. The man is wearing a hat. They are walking hand in hand on the leaf-covered road. Surrounded by lush green trees.
target image prompt: bitmoji style. The back view of an old couple. The man is wearing a hat. They are walking hand in hand on the leaf-covered road. Surrounded by lush green trees.

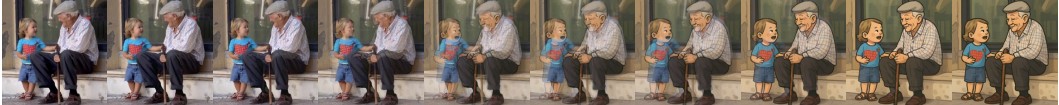

condition image prompt: A little boy holds the hand of an old man with a cane and they talk amiably.
target image prompt: bitmoji style. A little boy holds the hand of an old man with a cane and they talk amiably.

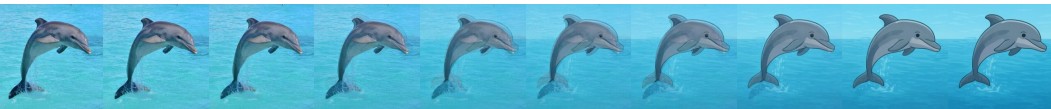

condition image prompt: Gray dolphin leaping in the sea.
target image prompt: bitmoji style. Gray dolphin leaping in the sea.

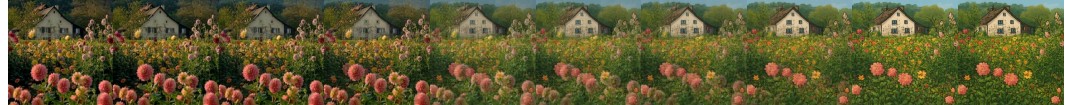

condition image prompt: Colorful flowers in the front, a light-colored house with a chimney in the back, surrounded by trees.
target image prompt: bitmoji style. Colorful flowers in the front, a light-colored house with a chimney in the back, surrounded by trees.

Figure 20: More style transfer generation results.

## D Failure Cases

While `DRA-Ctrl` successfully achieves controllable image generation in most cases, it may occasionally fail in the image-to-depth task, primarily manifesting as the presence of colored regions in the generated depth images. We attribute this limitation to the inherent nature of video models, which predominantly generate color data. A failure case is presented in Figure 21.

Origin Image      Generated Image

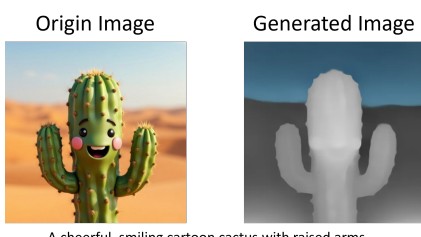

A cheerful, smiling cartoon cactus with raised arms
stands in a vibrant desert landscape under a blue sky

Figure 21: A failure case of `DRA-Ctrl`.

## E Societal Impact

Our work advances controllable image generation with significant societal implications, offering both opportunities for innovation and risks requiring proactive mitigation. Below, we outline the potential positive and negative impacts, alongside measures to address the latter.

On the positive side, our high-quality, controllable generation method empowers creative and practical applications. Artists and designers can leverage it to produce imaginative content efficiently, while

educators benefit from dynamically generated visual aids for teaching. The fine-grained control also enables ethical uses in journalism and advertising, enhancing productivity and accessibility across domains.

However, negative impacts must be acknowledged. Malicious actors could exploit the technology to create convincing fake images for disinformation, fraud, or impersonation; to mitigate this, we adopt a gated release of models to restrict access. Bias in training data might lead to stereotypical or discriminatory outputs, disproportionately harming marginalized groups — addressed through rigorous bias testing during development. Further, misuse for non-consensual imagery (e.g., deepfakes) necessitates monitoring mechanisms and legal safeguards to protect privacy.

In summary, while our technology unlocks creative and educational potential, its risks-particularly around misinformation, bias, and privacy-demand deliberate countermeasures. By combining technical safeguards with policy-oriented solutions, we aim to foster responsible use and maximize societal benefit.

## F   Safeguards

To mitigate potential misuse risks associated with our controllable image generation technology, we will implement a gated release strategy when making the models publicly available. This will include: comprehensive usage guidelines explicitly prohibiting malicious applications such as disinformation campaigns and non-consensual imagery generation; an access control mechanism requiring users to agree to ethical use terms before obtaining the model. While we recognize no safeguards can eliminate all risks, these measures represent our proactive commitment to responsible AI development and deployment.

