# OpenReview forum: "Dimension-Reduction Attack! Video Generative Models are Experts on Controllable Image Synthesis"
_NeurIPS.cc/2025/Conference — NeurIPS 2025 poster_

### Official Review · Reviewer_1JJJ · 2025-06-30

**Clarity:** 2
**Significance:** 2
**Originality:** 2
**Rating:** 4
**Confidence:** 4

**Summary:**

In this paper, the authors focus on the problem of image generation via a video generative model. They build on top of the HunyuanVideo-I2V model, training it with image pairs set as two frame videos with a mixup transition. They train their model on the Subjects200K and Text-to-Image-2M datasets and evaluate their approach on a number of diverse visual tasks: Canny to Image, Depth to Image, Deblurring, Colorization, and Masked out inpainting, and style transfer. They show quantative results on these tasks, showing improvement over baselines. Finally they perform an ablation analysis.

**Questions:**

1. Can you elaborate on the novelty of the approach? It seems that this is just finetuning a video model for image generation.

2. It isn't surprising that video priors would improve image generation. Could you elaborate more on the significance of this result?

**Ethical Concerns:**

["NO or VERY MINOR ethics concerns only"]

**Final Justification:**

The authors have addressed my concerns in the rebuttal. Considering also the comments of other reviewers, I keep the score at 4.

**Limitations:**

Yes, the authors have addressed the limitations of the work.

**Paper Formatting Concerns:**

I see no major formatting concerns apart from the unusual choice to put the figure before the abstract.

**Quality:**

3

**Strengths And Weaknesses:**

Strengths:

1. The paper focuses on a problem with immediate applications to computer graphics and computational photography: Image Generation

2. The paper shows promising results on a diverse number of tasks, not simply general image synthesis.

3. The approach is simple and makes intuitive sense: Video priors should strengthen image generation performance.

Weaknesses:

1. The paper's result is not particularly surprising. Video models probably outperform strict image models as the data domain of video is much richer.

2. The approach does not have many novel technical contributions, it's largely taking an existing video model and tuning it for better image generation performance.

3. There are some minor grammatical and stylistic issues (such as the choice to put a figure before the abstract), although these do not dramatically detract from readability.

---

> ### Author Rebuttal · Authors · 2025-07-30
>
> We are deeply grateful for the reviewer's positive assessment of our work, particularly immediate applications of our method, consistently strong performance across diverse tasks, and conceptually intuitive design. Below we provide comprehensive answers to the reviewer's questions.
>
> **Q1:** It is not surprising that video models outperform image model on controllable image generation tasks.
>
> **A1:**  We fully agree with the reviewer's perspective that video priors would significantly improve controllable image generation. Through our carefully designed mixup strategy, attention masking strategy, and FSPE, we have successfully transformed this intuitive concept into practical algorithms. The effectiveness of our method has been thoroughly validated through extensive experiments including comparative studies, ablation experiments, and qualitative evaluations on 7 different tasks.
>
> While the idea of using video priors to enhance controllable image generation may appear straightforward, we would like to clarify that effectively activating a model's prior knowledge for new tasks remains an important, challenging, and widely studied problem in computer vision research. This challenge has become particularly prominent given the abundance of powerful open-source models available in the community, making the alignment of pre-trained models to desired outcomes a highly active research area. Representative works in this direction include ControlNet [1], which leverages the priors of text-to-image generative models combined with external perceptual control conditions to accomplish new tasks, and CLIP [2], whose prior knowledge has been widely applied to image retrieval and semantic matching tasks, among others.
>
> In our work, the key research focus has been on how to effectively activate video priors. As detailed in Section 3.2 of our paper, **directly applying simple fine-tuning of T2V or I2V models leads to significant issues, either in terms of poor subject consistency or weak prompt adherence** (the quantitative experimental results are presented in Table 3 of our paper). To address these challenges in utilizing video priors, we proposed the mixup strategy that transforms the image generation problem into a very short video generation task through constructed transition frames. Simultaneously, we incorporated the attention masking strategy to ensure proper information interaction and FSPE to reduce the number of required transition frames, ultimately enhancing our method's performance. The significance of our work lies in its ability to effectively harness the inherent yet previously difficult-to-utilize prior knowledge in video models.
>
> **Q2:** Our method does not have many novel technical contributions.
>
> **A2:** We sincerely apologize for any confusion caused and will provide clarification on this matter. Our main technical contributions are summarized below.
>
> 1. Mixup strategy. Inspired by our observation that video generation models often transition between scenes using fade-in-fade-out effects, we propose a mixup strategy to construct transition frames, naturally transforming the controllable image generation task into a short video generation problem. To the best of our knowledge, we are the first to apply video models to controllable image generation via this mixup approach.
>
> 2. Novel attention masking strategy. To enhance the model's understanding of the condition image content, DRA-Ctrl occasionally incorporates condition image prompts. To ensure proper information flow, we introduce a novel attention masking strategy.
>
> 3. Frame Skip Position Embedding (FSPE). To reduce the number of required transition frames and improve efficiency, we propose FSPE, which enables a small number of frames to simulate longer sequences, achieving smooth transitions from the condition image to the target image.
>
> 4. VL score. For more accurate evaluation in subject-driven image generation tasks, we design a new metric leveraging powerful vision-language models.
>
> **Q3:** Some minor grammatical and stylistic issues.
>
> **A3:** We sincerely appreciate the reviewer's careful review, and we will address these grammatical and stylistic issues in the revised version.
>
> [1] Zhang, L., Rao, A., & Agrawala, M. (2023). Adding conditional control to text-to-image diffusion models. In Proceedings of the IEEE/CVF international conference on computer vision (pp. 3836-3847).
>
> [2] Radford, A., Kim, J. W., Hallacy, C., Ramesh, A., Goh, G., Agarwal, S., ... & Sutskever, I. (2021, July). Learning transferable visual models from natural language supervision. In International conference on machine learning (pp. 8748-8763). PmLR.

---

> > ### Comment · Area_Chair_RHxN · 2025-08-06
> >
> > I invite the reviewer to engage in the rebuttal discussion following author responses.

---

> > ### Comment · Reviewer_1JJJ · 2025-08-06
> >
> > Thank you for your response. The rebuttal has addressed some of my concerns.

---

> > > ### Author Response · Authors · 2025-08-07
> > >
> > > Dear Reviewer 1JJJ
> > >
> > > We are pleased to know that our responses have addressed your concerns. We are grateful for your precious time and constructive feedback.
> > >
> > > Best,
> > >
> > > DRA-Ctrl Authors

---

### Official Review · Reviewer_A4pM · 2025-07-02

**Clarity:** 3
**Significance:** 3
**Originality:** 3
**Rating:** 4
**Confidence:** 4

**Summary:**

The paper proposes a shift for the video generation models to use them for controllable image generation. In the paper, the framework, DRA-Ctrl, is proposed for task adaptation and knowledge compression. DRA-Ctrl employs two strategies: 1. Mixup-based transition to bridge the gap between videos and images and 2. Frame-Skip Position Embedding to allow large image transformations without generating many intermediate results. DRA-Ctrl is evaluated on various controllable image generation tasks such as subject-driven generation, style transferring, spatial-aligned controllable generation. The proposed model uses HunyuanVideo-I2V as backbone model.

**Questions:**

Please have a consistent style on the tables. Table 1 highlight the proposed model with yellow, but other tables don’t and they have DRA-Ctrl. In table 1, the best scores are in bold, while, in table 2 the best scores are highlighted red and the second-best scores are blue. For the rest of the tables the best score are in bold for VL results only. Moreover, some tables have arrow near name of metrics, some don’t. Please make tables consistent.

Thank you for your acknowledgment and detailed discussion on safeguards  and societal impact with both the potential positive and negative impacts of the proposed model!

**Ethical Concerns:**

["NO or VERY MINOR ethics concerns only"]

**Final Justification:**

I decided to maintain my score as Borderline Accept.

**Limitations:**

yes

**Quality:**

3

**Strengths And Weaknesses:**

Strengths:

1. Writing is clear and the flow is easy to follow.

2. The proposed model can be used for various condıtional image generation tasks.

3. The paper provides comprehensive evaluation and ablation studies.

Weaknesses:

1. Comparison with SD1.5 in Table 1 can be useless because it is the initial stable diffusion-based model and there are SDXL, SD2, SD3, and SD3.5. I find comparison with the recent models more interesting.

2. Possible long inference time: This is a known limitation of many diffusion-based image generation models, and it is likely a weakness of the proposed model as well. However, the paper provides limited analysis of inference time. While it includes an efficiency comparison with baseline models, this analysis should be extended to include state-of-the-art (SoTA) image editing and generation models. Without this broader comparison, it is difficult to be convinced of the proposed model’s efficiency.

---

> ### Author Rebuttal · Authors · 2025-07-30
>
> We thank the reviewer for recognizing our method's versatility across conditional image generation tasks, thorough experiments, and responsible discussion of safeguards and societal impact. Below we provide answers to the reviewer's questions.
>
> **Q1:** Comparison with the recent models such as SDXL, SD2, SD3, and SD3.5.
>
> **A1:** Thanks for this excellent point. In Table 1 in our paper, we focus our comparison on SD1.5 and FLUX-based methods because we consider FLUX's performance highly competitive with models like SDXL, SD3, etc. We completely agree that broader comparisons are essential, which is why we include them in Table 2, benchmarking against methods like MS-Diffusion (built on SDXL) and BootPIG (on SD2.1). The reviewer's feedback highlights that our presentation could be clearer. In the revised version, we will restructure our experimental tables to better integrate these results, ensuring the comprehensiveness of our evaluation is immediately apparent and avoids any misunderstanding.
>
> **Q2:** Possible long inference time.
>
> **A2:** We sincerely appreciate the reviewer's constructive suggestions. In response, we have supplemented our analysis with comparative results between DRA-Ctrl and other state-of-the-art (controllable) image generation methods, as presented in the Table 2. All experiments are conducted on a single H800 GPU with consistent settings: 50 inference steps for $512\times512$ image generation, evaluated on DreamBench. The results demonstrate that controllable image generation methods typically require around $2\times$ to $6\times$ inference time of their backbone models. Our DRA-Ctrl doubles the number of latent frames of UNO, a strong baseline (4 latent frames v.s. UNO's 2 latent frames), while maintaining reasonable computational overhead, the image generation time increases by only 63\% and demonstrating superior performance.
>
> Table 2: Generation speed of image generation models and controllable image generation methods.
> |  | Backbone | Generation Time/s | VL Score $\uparrow$ |
> | :--- | :---: | :---: | :---: |
> | SDXL | - | 2.15 | - |
> | FLUX | - | 4.09 | - |
> | MS-Diffusion | SDXL | **4.65** | 1.94 |
> | OminiControl | FLUX | 7.93 | 2.21 |
> | UNO | FLUX | 14.7 | 2.43 |
> | DRA-Ctrl | HunyuanVideo | 24.0 | **2.56** |
>
> **Q3:** Inconsistent style on the tables.
>
> **A3:** We sincerely appreciate the reviewer's careful review of our manuscript's formatting. In the revised version, we will implement the following improvements: 1) adopt a unified highlight color scheme throughout, 2) emphasize all optimal results by applying bold formatting (previously unbolded), and 3) consistently annotate all metrics with directional arrows for clarity.

---

> > ### Comment · Reviewer_A4pM · 2025-08-05
> > **Reply**
> >
> > Thank you for additional effort to address my concerns. I have read the response and other reviews carefully. I will keep my score as it is.

---

> > > ### Author Response · Authors · 2025-08-07
> > >
> > > Dear Reviewer A4pM
> > >
> > > We truly appreciate your feedback and the time you spent reviewing our work. It is encouraging to know that our clarifications have addressed your concerns.
> > >
> > > Best,
> > >
> > > DRA-Ctrl Authors

---

### Official Review · Reviewer_h3gK · 2025-07-02

**Clarity:** 3
**Significance:** 3
**Originality:** 3
**Rating:** 5
**Confidence:** 4

**Summary:**

This work targets at controllable image generation by leveraging existing video generation models. Multiple image generation is formulated into keyframe inbetween task in video generation. By carefully designning the keyframe transition between condition image and target one, the video capabilities in smooth transition over time are reused. Experimental results demonstrate the effectiveness of reusing video generative prior in image generation.

**Questions:**

- In terms of number of transition frames, what if we have no transition frames? That is, we directly feed the condition image and target one.
- As mentioned at L159, I2V model over-preserves similarity to the condition image and exhibits poor prompt adherence. After keyframe fine-tuning, is prompt following better?

**Ethical Concerns:**

["NO or VERY MINOR ethics concerns only"]

**Final Justification:**

Thanks for the response that addressed my concerns.

**Limitations:**

yes

**Quality:**

3

**Strengths And Weaknesses:**

This paper is well-organized and clearly presented.
The motivation for reusing the video generative prior is insightful for image applications.
The experimental details are thoroughly provides including training and evaluation, which helps reproduction.
The comparison against prior efforts is comprehensive. Ablations further help the audience better understand every components.

---

> ### Author Rebuttal · Authors · 2025-07-30
>
> We gratefully acknowledge the reviewer's positive assessment of our work, including conceptually insightful motivation for image applications and comprehensive experimental analysis. In response to the reviewer's questions, we address each point as follows.
>
> **Q1:** What if we have no transition frames?
>
> **A1:** We appreciate the reviewer's valuable suggestions for improving our work. We have conducted additional ablation studies on the number of transition frames, including an experiment without any transition frames. The results are presented in Table 1. The experimental results align with the I2V baseline findings: when no transition frames are used, the generated images maintain excessive similarity to the condition images (as evidenced by the high DINO and CLIP-I scores in the table), while demonstrating poor prompt adherence (indicated by the lower CLIP-T scores).
>
> Table 1: Ablation study on the number of transition frames.
> | number of transition frames | VL Score $\uparrow$ | DINO $\uparrow$ | CLIP-I $\uparrow$ | CLIP-T $\uparrow$ |
> | :---: | :---: | :---: | :---: | :---: |
> | 0 | 2.37 | 0.736 | **0.837** | 0.291 |
> | 4 | 2.19 | 0.692 | 0.820 | 0.292 |
> | 8 | **2.42** | **0.742** | 0.834 | **0.295** |
> | 12 | 2.09 | 0.715 | 0.826 | 0.283 |
>
> **Q2:** After finetuning, does the I2V model show improvement in 1) reducing excessive similarity between generated images and condition images, and 2) enhancing prompt adherence?
>
> **A2:** Yes. We clarify that the I2V model in L159 is derived from HunyuanVideo-I2v, finetuned using two-frame data from the Subjects200K dataset (referred to as I2V baseline in our paper). As demonstrated in Table 3 in the paper (shown in the following Table 2), I2V baseline exhibits two critical limitations: 1) excessive similarity to condition images, evidenced by very high DINO score surpassing even the oracle and very high CLIP-I score approaching oracle, and 2) poor prompt adherence, reflected in low CLIP-T score. However, under the same experimental setting, DRA-Ctrl exhibits high DINO and CLIP-I scores, yet significantly lower than I2V baseline, and improved CLIP-T score compared to I2V baseline. This result demonstrates that the keyframe-finetuned DRA-Ctrl avoids the two limitations of the I2V baseline.
>
> Table 2: Comparison with baselines.
> |  | VL Score $\uparrow$ | DINO $\uparrow$ | CLIP-I $\uparrow$ | CLIP-T $\uparrow$ |
> | :--- | :---: | :---: | :---: | :---: |
> | Oracle | - | 0.774 | 0.885 | - |
> | T2V Baseline | 2.01 | 0.658 | 0.787 | **0.306** |
> | I2V Baseline | 2.34 | **0.803** | **0.874** | 0.291 |
> | DRA-Ctrl | **2.44** | 0.715 | 0.821 | 0.298 |

---

### Official Review · Reviewer_MS8a · 2025-07-03

**Clarity:** 3
**Significance:** 3
**Originality:** 3
**Rating:** 4
**Confidence:** 3

**Summary:**

This work proposes using video generation models (e.g., Hunyuan Video) to achieve spatial structure-controlled image generation.
A mix-based transition strategy is introduced to bridge the gap between videos and images.
The authors also propose a Frame-Skip Position Embedding, which expands temporal intervals in the latent space.
Experiments are conducted under various types of conditions to verify the effectiveness of the proposed approach.

**Questions:**

How does the image generation speed compare to that of Flux and SD?

**Ethical Concerns:**

["NO or VERY MINOR ethics concerns only"]

**Final Justification:**

Thanks for the author's response. Most of my concerns are resolved and I will maintain my initial score as borderline accept.

**Limitations:**

yes

**Paper Formatting Concerns:**

The format is generally good

**Quality:**

3

**Strengths And Weaknesses:**

Weaknesses
The generation quality is sometimes worse than that of Flux, but I think it is acceptable.
I wonder whether the inference speed is significantly slower when using video models for image generation.
The authors are encouraged to provide an analysis of why using video models can greatly improve controllability, and to include any failure cases where the method struggles to provide effective control.

Strengths
The improvement in controllability is impressive.
The topic explores an interesting and promising direction.
The experiments are comprehensive.

---

> ### Author Rebuttal · Authors · 2025-07-30
>
> We sincerely appreciate the reviewer's positive feedback on our work, particularly the recognition of DRA-Ctrl's exceptional controllability, the intriguing and promising nature of our research topic, and the thoroughness of our experiments. Regarding the reviewer's questions, we make the following responses.
>
> **Q1:** The generation quality is sometimes worse than that of Flux, though acceptable.
>
> **A1:** Thanks for the comment. We fully agree that visual quality is an important evaluation dimension. The slight degradation observed in some cases results from our use of HunyuanVideo as the backbone, which does not match Flux in overall visual performance. We chose it deliberately to ensure a fair and controlled comparison, as it shares architectural similarities with Flux (both based on the MM-DiT family) and was state-of-the-art at the time of submission. This setup allows us to isolate the effects of our proposed paradigm and architectural innovations without conflating them with differences in backbone design. While newer models like WAN 2.2 offer better visual quality, their architectures differ significantly from Flux, making them less suitable for direct comparison. we have observed further improvements when applying our method to such models, and we will continue to adopt the strongest open-source backbones and contribute our advances to the community to support broader progress in this area.
>
> **Q2:** Generation speed compared to image generation model and other controllable image generation methods.
>
> **A2:** We thank the reviewer for the highly constructive comments. We evaluate the average single-image generation time on a single H800 GPU with 50 inference steps for generating $512\times512$ images on DreamBench, and the results are presented in Table 1. The results demonstrate that controllable image generation methods typically require around $2\times$ to $6\times$ inference time of their backbone models. Our DRA-Ctrl doubles the number of latent frames of UNO, a strong baseline (4 latent frames v.s. UNO's 2 latent frames), while maintaining reasonable computational overhead, the image generation time increases by only 63% and demonstrating superior performance.
>
> Table 1: Generation speed of image generation models and controllable image generation methods.
> |  | Backbone | Generation Time/s | VL Score $\uparrow$ |
> | :--- | :---: | :---: | :---: |
> | SDXL | - | 2.15 | - |
> | FLUX | - | 4.09 | - |
> | MS-Diffusion | SDXL | **4.65** | 1.94 |
> | OminiControl | FLUX | 7.93 | 2.21 |
> | UNO | FLUX | 14.7 | 2.43 |
> | DRA-Ctrl | HunyuanVideo | 24.0 | **2.56** |
>
> **Q3:** Analysis of why video models can improve controllability. Failure cases of DRA-Ctrl.
>
> **A3:** Thanks for the comment. Video models are trained on a large number of frames with spatiotemporal dependencies, thereby encoding rich prior knowledge about long-range context, consistent object transitions, non-rigid transformations, and high-level scene dynamics. These priors align well with goals of controllable image generation, as discussed in L32-35 in our paper. DRA-Ctrl successfully leverages these inherent priors from video models for controllable image generation, achieving superior performance. In contrast, image models are primarily trained with text-image pairs and lack such prior knowledge relevant to controllable image generation. They struggle to understand object pose variations, background changes, and other image transformations. Therefore, adopting video models as the backbone demonstrates clear advantages over using image models. Furthermore, we visualized the attention maps of DRA-Ctrl and observed that during subject-driven image generation, the attention weights between the final frame and the condition image are significantly higher in the subject regions. This clearly demonstrates DRA-Ctrl's strong controllability in preserving subject fidelity.
>
> While DRA-Ctrl successfully achieves controllable image generation in most cases, it may occasionally fail in the image-to-depth task, primarily manifesting as the presence of colored regions in the generated depth images. We attribute this limitation to the inherent nature of video models, which predominantly generate color data. The limited LoRA fine-tuning appears insufficient to completely prevent the model from producing partial color outputs in these cases. We have included these failure cases in the revised version.

---

### Official Review · Reviewer_hSh7 · 2025-07-05

**Clarity:** 3
**Significance:** 4
**Originality:** 2
**Rating:** 5
**Confidence:** 4

**Summary:**

- This work proposes Dimension-Reduction Attack (DRA-Ctrl), a method that utilizes the long-range context modeling and full attention in video generative models to perform various controllable image generation tasks.
- Specifically, authors leverage priors from Image-to-Video (I2V) models for the tasks of 1) subject-driven generation, image-to-image translation and style transfer, and demonstrate that such repurposed video models outperform those trained only on images.
- The main idea of the paper is to construct a synthetic video with the condition image as the first frame and the target image as its last frame. The intermediate frames are generated using a temporal position-aware mixup strategy between the first and the last frames. An I2V model is then trained on these synthetic videos to solve several image-to-image translation tasks. To avoid computation costs due to processing a lot of intermediate frames, they propose Frame-skip position embedding (FSPE) to augment the 3D RoPE embeddings. FSPE "simulates" a longer video in the latent space by expanding the PEs.
- Experiments demonstrate that video generative models can be repurposed to solve controllable image generation tasks and consistently outperform methods built upon image generation tasks

**Questions:**

- Why is the method called dimension-reduction "attack"? What is the attack that is being done in the proposed method? The rest of the title conveys the workings of the proposed method but the name is very misleading.
- For the VL score, how do you ensure that the model returns the same score for a particular image pair in a randomized validation set? In my experience a VLM outputs different scores for the same input when asked at a later point in time. This makes the metric very unreliable.
- What is the Oracle score in Table 2? If it is computed using the ground truth images, why can't the VL-Score metric be computed for this row?
- Please comment on my issue with the experimental setup described in the weaknesses section.

**Ethical Concerns:**

["NO or VERY MINOR ethics concerns only"]

**Final Justification:**

I thank the authors for their thorough rebuttal. Authors have satisfactorily answered my questions about computational cost, VLM output stability and the oracle score. AFter reading through other reviews and author response, I vote to keep my initial score.

**Limitations:**

Yes

**Quality:**

3

**Strengths And Weaknesses:**

**Strengths**
- The paper is well written and easy to follow. The paper has adequate details for fair reproduction.
- The results on image-to-image translation (Table-1) and subject-driven generation are impressive.
- The use of mixup, a data augmentation strategy widely used in self-supervised learning, for media generation is novel and this line of work can potentially improve the "mixup" of ideas from both the domains.
- The idea that leveraging video generation models for improved controlled image generation tasks is very interesting and can potentially open up several valuable research problems.

**Weaknesses**
- My main issue with this work is the evaluation setup. Evaluation setup compares DRA-Ctrl with contemporary work. It fails to normalize the results with compute. A more fair way to evaluate DRA-Ctrl would be to report FLOPs and GPU hours spent by the pre-training models and the finetuning step and then comparing results.
- Here is the setup that I believe is fair. 1) Image vs Video pre-trained model initialization: Compare a method that uses an image generation model's (trained in N gpu hours, M Flops) weights as initialization to DRA-Ctrl, that initializes from a video generation model trained with the same GPU hours and FLOPs. This experiment would be convincing that the priors learned by video generation model can indeed improve image generation. With the current setup it is not clear if the compute budget of pre-trained model or the video generation task itself is helping. 2) Number of parameters/compute budget in the fine-tuning stage: Compare with methods that use similar compute budgets/parameters in the 2nd stage to fully understand where the improvements are coming from.

---

> ### Author Rebuttal · Authors · 2025-07-30
>
> We sincerely appreciate the reviewers' recognition of our work, particularly the acknowledgment that the mixup strategy in DRA-Ctrl is novel and extensive, as well as the assessment that our approach is interesting and has the potential to open up new research directions. We are also deeply grateful for the highly professional and insightful questions raised by the reviewers. Below, we address each of these points in detail.
>
> **Q1:** Concerns that the differences in computational costs may lead to an unfair comparison, when comparing the video model-based method (DRA-Ctrl) and image model-based methods.
>
> **A1:** We sincerely appreciate the reviewer's insightful observations regarding the evaluation setup. We have taken note of these concerns during our experiments and have made every effort to align the evaluation setup when comparing these two categories of methods.
>
> 1) About pre-trained models. The reviewer’s point about controlling computational budgets is well taken. Matching resources between the pretrained image and video models would clarify if DRA-Ctrl’s advantage comes from extra compute or the video task itself. However, video generation is inherently more complex than image generation, which naturally requires more GPUs and time. Due to limited resources, strict alignment during pretraining was not feasible. To address this, we chose HunyuanVideo as our backbone because its architecture closely matches FLUX. This choice helps us align the pretrained models as much as possible.
>
> 2) About number of parameters/compute budget in the fine-tuning stage. Since UNO uses FLUX as its backbone, we compare computational budget between DRA-Ctrl and UNO. As demonstrated in Table 1, DRA-Ctrl achieves better performance with less computational budget.
>
> Table 1: Computational budget comparison between DRA-Ctrl and UNO.
> |  | DRA-Ctrl | UNO |
> | :--- | :---: | :---: |
> | number of parameters | 92 M | 956 M |
> | total batch size | 32 | 16 |
> | training steps | 9 k | 10 k |
> | VL Score $\uparrow$ | 2.56 | 2.43 |
>
> **Q2:** The meaning behind the name (DRA-Ctrl) of our method.
>
> **A2:** We apologize for any confusion our method's naming may have caused the reviewer. The name is inspired by the science fiction novel *The Three-Body Problem*, which introduces "Dimension-Reduction Attack", a devastating attack employed by advanced civilizations against lesser ones, characterized by forcibly reducing three-dimensional space to two dimensions with overwhelming power.
>
> In our work, we leverage a 3D video generation model to tackle 2D image generation tasks, achieving superior performance. Drawing an analogy to the novel’s concept, we refer to our approach as DRA-Ctrl to highlight its capability to bridge and dominate across dimensions.
>
> **Q3:** Concerns about VLM output instability.
>
> **A3:** The reviewer raises a valid observation regarding the inherent variability in VLM outputs. Our VL Score remains statistically reliable for evaluation purposes, supported by the following evidence. 1) Our assessment is conducted on DreamBench, which requires evaluating $30\times25=750$ images per method and averaging the results. This extensive sampling significantly reduces the impact of random fluctuations. 2) We systematically tested each method with random seeds 0-4 (5 independent runs). As shown in Table 2, while absolute scores show minor variations across seeds, the relative ranking between methods remains consistently stable within each seed. 3) Furthermore, the use of VLM for evaluation has been widely adopted in recent literature[1][2].
>
> Table 2: VL Scores across different random seeds.
> | Random Seed | 0 (rank) | 1 (rank) | 2 (rank) | 3 (rank) | 4 (rank) | Average (rank) |
> | :--- | :--- | :--- | :--- | :--- | :--- | :---: |
> | OminiControl | 2.21 (3) | 2.21 (3) | 1.98 (3) | 2.50 (3) | 2.51 (3) | 2.28 (3) |
> | MS-Diffusion | 1.94 (4) | 1.94 (4) | 1.75 (4) | 2.27 (4) | 2.30 (4) | 2.04 (4) |
> | UNO | 2.43 (2) | 2.43 (2) | 2.13 (2) | 2.73 (2) | 2.75 (2) | 2.49 (2) |
> | DRA-Ctrl | **2.56** (1) | **2.56** (1) | **2.23** (1) | **2.83** (1) | **2.82** (1) | **2.60** (1) |
>
> **Q4:** The Oracle score of VL Score.
>
> **A4:** Thanks for the comment. Unlike DINO and CLIP-I metrics, which only require image pairs [3], computing the VL Score also needs prompt information. Since DreamBench does not provide ground-truth images for every subject under each prompt, we do not report oracle values for the VL Score in our paper. However, to establish a meaningful benchmark, we manually annotate prompts for each image in DreamBench. For each subject, we select one image as the reference and treat the others as ground truth. This enables us to compute a provisional VL Score oracle, as shown in Table 3.
>
> Table 3: The oracle score of VL Score.
> |  | Random Seed = 0 | Average over Random Seeds 0-4 |
> | :--- | :---: | :---: |
> | Oracle | 2.61 | 2.60 |
> | OminiControl | 2.21 | 2.28 |
> | MS-Diffusion | 1.94 | 2.04 |
> | UNO | 2.43 | 2.49 |
> | DRA-Ctrl | **2.56** | **2.60** |
>
> [1] Zhuang, C., Huang, A., Cheng, W., Wu, J., Hu, Y., Liao, J., ... \& Zhang, C. (2025). Vistorybench: Comprehensive benchmark suite for story visualization. arXiv preprint arXiv:2505.24862.
>
> [2] Tan, Z., Liu, S., Yang, X., Xue, Q., \& Wang, X. (2024). Ominicontrol: Minimal and universal control for diffusion transformer. arXiv preprint arXiv:2411.15098.
>
> [3] Ruiz, N., Li, Y., Jampani, V., Pritch, Y., Rubinstein, M., \& Aberman, K. (2023). Dreambooth: Fine tuning text-to-image diffusion models for subject-driven generation. In Proceedings of the IEEE/CVF conference on computer vision and pattern recognition (pp. 22500-22510).

---

### Comment · Area_Chair_RHxN · 2025-08-04
**Discussion engagement**

Hello all. Thanks to all reviewers and authors for their implication in the review process so far. If not already done, we encourage all reviewers to engage with authors following rebuttals since the discussion period is coming to a close in a few days. Thank you for helping ensure a quality  NeurIPS review process!

---

### Note · Authors · 2025-08-12

We sincerely appreciate all reviewers and the AC for their thorough evaluation and valuable suggestions on our work. We are greatly encouraged that DRA-Ctrl has received consistently positive ratings from all reviewers. Specifically, reviewers find DRA-Ctrl to be:

- Intuitive, novel, and interesting in motivation and methodology
- Having the potential to open new research directions
- Comprehensive in experimental design
- Outstanding and effective in performance

During the discussion period, we have addressed the concerns raised by reviewers:

- **​Fair comparison**: We conducted detailed analysis of DRA-Ctrl's computational overhead and demonstrated that DRA-Ctrl achieves superior performance with lower computational overhead compared to UNO.
- **Evaluation based on Vision-language models**: Through additional experiments, we confirmed the stability of VLM outputs and provided a reference oracle score.
- **Generation speed**: We added comprehensive comparisons with other controllable image generation methods and image generation models, demonstrating that the generation speed of DRA-Ctrl is reasonable and DRA-Ctrl achieves superior performance.
- **​Ablation studies**: We included a new experiment without transition frames, conclusively proving their effectiveness in improving prompt adherence and reducing image over-similarity.
- **​Technical contributions**: We re-clarified DRA-Ctrl's technical innovations and further explained why directly using video generative models is infeasible, emphasizing the necessity of our approach.
- ​**Other details**: We elaborated on the naming rationale, analyzed controllability enhancement mechanisms, and will ensure consistent table formatting while addressing all grammatical/stylistic issues in the new version.

While we understand this concludes our interaction, we hope our responses have comprehensively addressed all concerns. We respectfully ask reviewers and the AC to evaluate whether our clarifications and proposed revisions adequately resolve the raised issues. We remain fully committed to implementing all suggested improvements in the final version.

We sincerely appreciate your time and thoughtful engagement throughout this review process.

---

### Decision · Program_Chairs · 2025-09-17

**Decision:**

Accept (poster)

**Comment:**

The paper proposes a dimension-reduction-based approach for control in video generative models. The reviewers found the method to be sound and to improve on existing approaches.